# Equivariant Disentangled Transformation for Domain Generalization under Combination Shift

## Abstract

Machine learning systems may encounter unexpected problems when the data distribution changes in the deployment environment. A major reason is that certain combinations of domains and labels are not observed during training but appear in the test environment. Although various invariance-based algorithms can be applied, we find that the performance gain is often marginal. To formally analyze this issue, we provide a unique algebraic formulation of the *combination shift* problem based on the concepts of homomorphism, equivariance, and a refined definition of disentanglement. The algebraic requirements naturally derive a simple yet effective method, referred to as *equivariant disentangled transformation* (EDT), which augments the data based on the algebraic structures of labels and makes the transformation satisfy the equivariance and disentanglement requirements. Experimental results demonstrate that invariance may be insufficient, and it is important to exploit the equivariance structure in the combination shift problem.

## 1 Introduction

The way we humans perceive the world is *combinatorial* — we tend to cognize a complex object or phenomenon as a combination of simpler factors of variation. Further, we have the ability to recognize, imagine, and process novel combinations of factors that we have never observed so that we can survive in this rapidly changing world. Such ability is usually referred to as *generalization*. However, despite recent super-human performance on certain tasks, machine learning systems still lack this generalization ability, especially when only a limited subset of all combinations of factors are observable (Sagawa et al., 2020; Träuble et al., 2021; Goel et al., 2021; Wiles et al., 2022). In risk-sensitive applications such as driver-assistance systems (Alcorn et al., 2019; Volk et al., 2019) and computer-aided medical diagnosis (Castro et al., 2020; Bissoto et al., 2020), performing well only on a given subset of combinations but not on unobserved combinations may cause unexpected and catastrophic failures in a deployment environment.

*Domain generalization* (Wang et al., 2021a) is a problem where we need to deal with combinations of two factors: domains and labels. Recently, Gulrajani & Lopez-Paz (2021) questioned the progress of the domain generalization research, claiming that several algorithms are not significantly superior to an empirical risk minimization (ERM) baseline. In addition to the model selection issue raised by Gulrajani & Lopez-Paz (2021), we conjecture that this is due to the ambitious goal of the usual domain generalization setting: generalizing to a completely unknown domain. Is it really possible to understand art if we have only seen photographs (Li et al., 2017)? Besides, those datasets used for evaluation usually have almost uniformly distributed domains and classes for training, which may be unrealistic to expect in real-world applications.

A more practical but still challenging learning problem is to learn all domains and labels, but only given a limited subset of the domain-label combinations for training. We refer to the usual setting of domain generalization as *domain shift* and this new setting as *combination shift*. An illustration is given in Fig. 1. Combination shift is more feasible because all domains are at least partially observable during training but is also more challenging because the distribution of labels can vary significantly across domains. The learning goal is to improve generalization with as few combinations as possible.

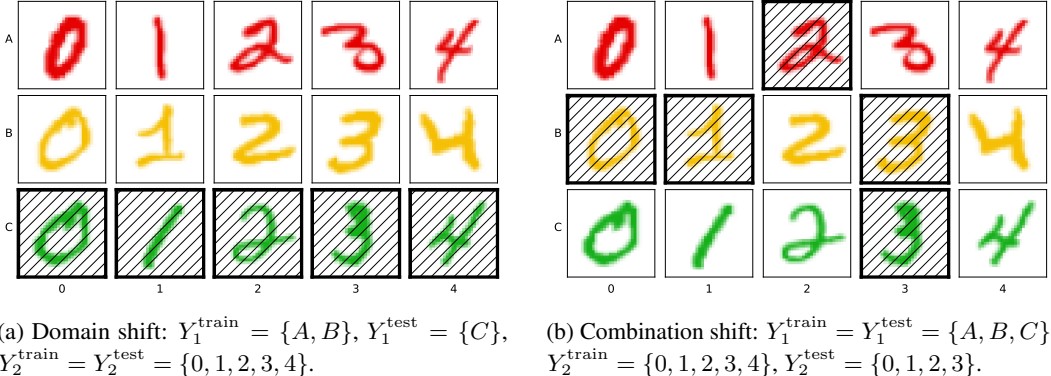

(a) Domain shift: $Y_1^{\text{train}} = \{A, B\}$, $Y_1^{\text{test}} = \{C\}$, $Y_2^{\text{train}} = Y_2^{\text{test}} = \{0, 1, 2, 3, 4\}$.

(b) Combination shift: $Y_1^{\text{train}} = Y_1^{\text{test}} = \{A, B, C\}$, $Y_2^{\text{train}} = \{0, 1, 2, 3, 4\}$, $Y_2^{\text{test}} = \{0, 1, 2, 3\}$.

Figure 1: Domain generalization under *domain shift* (an unseen domain) and *combination shift* (unseen combinations of domains and labels). Domain: color, label: digit, training: □, test: ▨.

To solve the combination shift problem, a straightforward way is to apply the methods designed for domain shift. One approach is based on the idea that the prediction of labels should be invariant to the change of domains (Ganin et al., 2016; Sun & Saenko, 2016; Arjovsky et al., 2019; Creager et al., 2021). However, we find that the performance improvement is often marginal. Recent works (Wiles et al., 2022; Schott et al., 2022) also provided empirical evidence showing that invariance-based domain generalization methods offer limited improvement. On the other hand, they also showed that data augmentation and pre-training could be more effective. To analyze this phenomenon, a unified perspective on different methods is desired.

In this work, we provide an *algebraic formulation* for both invariance-based methods and data augmentation methods to investigate why invariance may be insufficient and how we should learn data augmentations. We also derive a simple yet effective method from the algebraic requirements, referred to as *equivariant disentangled transformation* (EDT), to demonstrate its usefulness.

Our main contributions are as follows:

- We provide an algebraic formulation for the *combination shift* problem. We show that *invariance* is only half the story and it is important to exploit the *equivariance* structure. We present a refined definition of *disentanglement* beyond the one based on group action (Higgins et al., 2018), which may be interesting in its own right.

- Based on this algebraic formulation, we derive (a) what *combinations* are needed to effectively learn augmentations; (b) what *augmentations* are useful for improving generalization; and (c) what *regularization* can be derived from the algebraic constraints, which can serve as a guidance for designing data augmentation methods.

- As a proof of concept, we demonstrate that *learning data augmentations based on the algebraic structures of labels* is a promising approach for the combination shift problem.

## 2 PROBLEM: DOMAIN GENERALIZATION UNDER COMBINATION SHIFT

Throughout the following sections, we study the problem of transforming a set of features $X$ to a set of targets $Y$ via a function $f : X \to Y$. Here, $X$ can be a set of images, texts, audios, or more structured data, while $Y$ is the space of outputs. Further, the target $Y$ may have multiple components. For example, $Y_1$ is the set of domain indices and $Y_2$ is the set of target labels.

Ideally, all combinations of domains and target labels would be uniformly observable. However, in reality, it may not be the case because of selection bias, uncontrolled variables, or changing environments (Sagawa et al., 2020; Träuble et al., 2021). Let $Y_i^{\text{train}}$ and $Y_i^{\text{test}}$ denote the sets of $i$-th components (the support of the marginal distributions) observed in the training and test data. In the usual domain generalization setting (Wang et al., 2021a; Gulrajani & Lopez-Paz, 2021), the goal is to generalize to a completely unseen domain, i.e., domain shift. We have $Y_2^{\text{train}} = Y_2^{\text{test}}$ but

$Y_1^{\text{train}} \cap Y_1^{\text{test}} = \varnothing$. However, it is unclear how different domains should relate and why a model can generalize without the knowledge of the unknown domain (Wiles et al., 2022).

In this work, we focus on a more practical condition, called *combination shift* and illustrated in Fig. 1, where all test domains and labels can be observed separately during training, i.e., $Y_i^{\text{test}} \subseteq Y_i^{\text{train}} (i = 1, 2)$, but not all their combinations. An example is the spurious relationship problem (Torralba & Efros, 2011), such as the co-occurrence of the objects and their background (Sagawa et al., 2020). In an extreme case, the combinations in the training and test sets could be disjoint, which requires completely out-of-distribution generalization. We survey related problems and approaches in more detail in Appendix D.

## 3 FORMULATION: EQUIVARIANCE TO PRODUCT ALGEBRA ACTIONS

This section outlines the concepts needed to formally describe the problem and our proposed method. See Appendices B and C for a more detailed review and concrete examples. Those who are interested in the proposed method itself may skip this section and directly jump to Section 4.

Because in the domain generalization problem, we have at least two sets, domains and labels, it is natural to study their *product* structure, which is manifested as statistical *independence* or operational *disentanglement*. We focus on the latter and use the following definition:

**Definition 1.** Let $\{\mathbf{A}_i = (A_i, \{f_i^j : A_i^{n_j} \to A_i\}_{j \in J_i})\}_{i \in I}$ be algebras indexed by $i \in I$, each of which consists of the underlying set $A_i$ and a collection of operations $f_i^j$ of arity $n_j$ indexed by $j \in J_i$. Let $\mathbf{A} = \prod_{i \in I} \mathbf{A}_i$ be the product algebra whose underlying set is the product set $A = \prod_{i \in I} A_i$. Let $\mathbf{A}$ act on sets $X$ and $Y$ via actions $\text{act}_X : A \times X \to X$ and $\text{act}_Y : A \times Y \to Y$. A transformation $f : X \to Y$ is disentangled if it is equivariant to $\text{act}_X$ and $\text{act}_Y$.

In short, a disentangled transformation is a function *equivariant to actions by a product algebra*. Note that a definition of disentangled representations based on product group action has been given in Higgins et al. (2018), which is a special case when $\{\mathbf{A}_i\}_{i \in I}$ are all groups. We emphasize that the concept of *disentanglement* is rooted in *product*, not group nor action. We will unwind this definition and discuss the reasons for this extension as well as its limitations below.

### 3.1 HOMOMORPHISM AND EQUIVARIANCE

An **algebra** consists of one or more *sets*, a collection of *operations* on these sets, and a collection of universally quantified equational *axioms* that these operations need to satisfy. A **homomorphism** between algebras is a function between the underlying sets that preserves the algebraic structure.

A (left) **action** of a set $A$ on another set $X$ is simply a binary function $\text{act} : A \times X \to X$. An action is equivalent to its *exponential transpose* or *currying*, a function $\widehat{\text{act}} : A \to X^X$ from $A$ to the set of endofunctions $X^X$, also known as a **representation** of $A$ on $X$. An action is *faithful* if all endofunctions are distinct, and *trivial* if all elements are mapped to the identity function $\text{id}_X$.

Let $\text{act}_X$ and $\text{act}_Y$ be actions of $A$ on $X$ and $Y$, respectively. A function $f : X \to Y$ is **equivariant** to $\text{act}_X$ and $\text{act}_Y$ if

$$\forall a \in A, f \circ \widehat{\text{act}}_X(a) = \widehat{\text{act}}_Y(a) \circ f. \tag{1}$$

Specifically, if $\text{act}_Y$ is trivial, $f$ is called **invariant** to $\text{act}_X$:

$$\forall a \in A, f \circ \widehat{\text{act}}_X(a) = f. \tag{2}$$

In summary, for an underlying set $X$, an algebra over $X$ describes the structure of the set $X$ itself, while an action or a representation of another algebraic structure $A$ on $X$ describes the structure of a subset of the endofunctions $X^X$. Homomorphisms and equivariant functions describe how the structures of the set and endofunctions are preserved, respectively. An equivariant map can be also considered as a homomorphism between two algebras whose operations are all unary and indexed by elements in the set $A$. Note that only the equivariance — the structure of endofunctions — may not fully characterizes a learning problem, because not all operations are unary operations. In some problems, it would be necessary to consider the preservation of the structure of other operations with the concept of algebra homomorphism. See also Appendices A to C.

## 3.2 MONOID AND GROUP

Let us focus on the endofunctions $X^X$ for now. A way to describe the structure of a subset of endofunctions $X^X$ is to specify an algebra $\mathbf{A}$ and an action of $\mathbf{A}$ on $X$ preserving the algebraic structure. For example, an important operation is the **function composition** $\circ : X^X \times X^X \to X^X$, which can be described by how an action preserves a binary operation $\cdot : A \times A \to A$:

$$\forall a_1, a_2 \in A, \widehat{\mathrm{act}}(a_1 \cdot a_2) = \widehat{\mathrm{act}}(a_1) \circ \widehat{\mathrm{act}}(a_2). \tag{3}$$

Since the function composition is associative, $(A, \cdot)$ should be a *semigroup*. If we also want to include the **identity function** $\mathrm{id}_X$, then there should exist an *identity element* $e \in A$ (a nullary operation), which makes $(A, \cdot, e)$ a *monoid*.

*Remark* 1 (Group). If we only consider invertible endofunctions, then $A$ becomes a *group* (Higgins et al., 2018). However, only considering groups could be too restrictive. For example, periodic boundary conditions are required (Higgins et al., 2018; Caselles-Dupré et al., 2019; Quessard et al., 2020; Painter et al., 2020) for two-dimensional environments (e.g., dSprites (Matthey et al., 2017)), so that all the movements are invertible and have a cyclic group structure. This is only possible in synthetic environments such as games, not in the real world. Another example is the 3D Shapes dataset (Burgess & Kim, 2018), which consists of images of three-dimensional objects with different shapes, colors, orientations, and sizes. It is acceptable to model the shape, color, and orientation with permutation groups or cyclic groups. However, it is unreasonable if we increase the size of the largest object, then it becomes the smallest. This is because we only consider the set of *natural numbers*, representing size, count, or price, and of which addition only has a monoid structure. Therefore, it is important to consider endofunctions in general, not only the invertible ones. In this work, we mainly focus on monoid actions that only describe the function composition and identity function.

## 3.3 PRODUCT AND DISENTANGLEMENT

Finally, we are in a position to introduce the concept of disentanglement used in Definition 1. For two objects $Y_1$ and $Y_2$, we can consider their **product** $Y = Y_1 \times Y_2$, which is defined via a pair of canonical **projections** $p_1 : Y_1 \times Y_2 \to Y_1$ and $p_2 : Y_1 \times Y_2 \to Y_2$. This means that we can divide the product into parts and process each part separately without losing information. We reiterate that:

*Product structure is the core of disentanglement.*

Specifically, (a) if $Y_1$ and $Y_2$ are just sets, $Y$ is their *Cartesian product*; (b) if $Y_1$ and $Y_2$ have algebraic structures, $Y$ is the *product algebra* and the operations are defined componentwise; and (c) if $\mathbf{A}_1$ and $\mathbf{A}_2$ act on $Y_1$ and $Y_2$, respectively, then the product algebra $\mathbf{A} = \mathbf{A}_1 \times \mathbf{A}_2$ can act on $Y = Y_1 \times Y_2$ componentwise.

Additionally, if we let $PY$ be the set of all measures on $Y$, then $PY_1 \times PY_2$ is the set of joint distributions where two components are *statistically independent*, while $PY = P(Y_1 \times Y_2)$ is the set of all possible joint distributions. Product is the common denominator for all the definitions of disentanglement. In Definition 1, we only considered the product structure of endofunctions.

We can use this definition to formulate the domain generalization problem as follows. We assume that $Y = Y_1 \times Y_2$ has two components, where $Y_1$ is the set of domain indices and $Y_2$ is the set of other target labels. we choose a structure of a subset of the endofunctions $Y^Y$, described by two algebras $\mathbf{A}_1$ and $\mathbf{A}_2$ and two actions $\mathrm{act}_{Y_1}$ and $\mathrm{act}_{Y_2}$. Then, we let the product algebra $\mathbf{A} = \mathbf{A}_1 \times \mathbf{A}_2$ act on $Y = Y_1 \times Y_2$ componentwise via an action $\mathrm{act}_Y$. We also assume that there is an action $\mathrm{act}_X$ of $\mathbf{A}$ on $X$ that manipulates the features. After properly choosing the algebras and actions, the problem can be then formulated as finding a function equivariant to $\mathrm{act}_X$ and $\mathrm{act}_Y$.

Note that it is usually unnecessary and sometimes impossible to decompose $X$ into a product, i.e., $X = X_1 \times X_2$ may not exist. For example, when $X$ is a set of objects with different shapes and colors, there does not exist an object without color. In this case, we could only equip the endofunctions $X^X$ with a product structure.

# 4 METHOD: EQUIVARIANT DISENTANGLED TRANSFORMATION

In this section, we present our proposed method based on an algebraic formulation of the combination shift problem. The basic idea is that if we choose the algebra properly, the algebraic requirements of the transformation naturally lead to useful architectures and regularization.

In the following discussion, we assume that $\mathrm{act}_{Y_i}(a_i, y_i) = y_i'$ for some $a_i \in A_i$ and $y_i, y_i' \in Y_i$, $i = 1, 2$. We denote an instance whose labels are $y_1$ and $y_2$ by $x_{y_1, y_2}$.

## 4.1 MONOID STRUCTURE

First, we discuss how to choose the algebra that is suitable for our problem and derive the algebraic requirements. As discussed in Section 3.2, we only require that algebras $\mathbf{A}_1$ and $\mathbf{A}_2$ are *monoids*, which means that there exist associative binary operations $\cdot_i : A_i \times A_i \to A_i$ and identity elements $e_i \in A_i$ for $i = 1, 2$. Then, according to Eq. (3) (*action commutes with composition*), we can derive that a product action $\widehat{\mathrm{act}}(a_1, a_2)$ on $X$ or $Y$ can be decomposed in two ways:

$$\widehat{\mathrm{act}}(a_1, a_2) = \widehat{\mathrm{act}}(a_1, e_2) \circ \widehat{\mathrm{act}}(e_1, a_2) = \widehat{\mathrm{act}}(e_1, a_2) \circ \widehat{\mathrm{act}}(a_1, e_2). \tag{4}$$

Or equivalently, the following diagram commutes (when the action is on $Y = Y_1 \times Y_2$):

$$
\begin{array}{ccc}
(y_1, y_2) & \xrightarrow{\widehat{\mathrm{act}}(e_1, a_2)} & (y_1, y_2') \\
\widehat{\mathrm{act}}(a_1, e_2) \downarrow & \quad \widehat{\mathrm{act}}(a_1, a_2) & \downarrow \widehat{\mathrm{act}}(a_1, e_2) \\
(y_1', y_2) & \xrightarrow{\widehat{\mathrm{act}}(e_1, a_2)} & (y_1', y_2')
\end{array}
\tag{5}
$$

Thus, we can focus on the endofunctions of the form $\widehat{\mathrm{act}}(a_1, e_2)$ and $\widehat{\mathrm{act}}(e_1, a_2)$, whose compositions constitute all endofunctions of interest.

*Remark* 2 (Size). Denoting the cardinality of a set $A$ by $|A|$ and the image of a function $f$ on a set $X$ by $f[X]$, i.e., a set defined by $\{f(x) \mid x \in X\}$, we can prove that $|\widehat{\mathrm{act}}([A_1], e_2)| \leq |A_1|$, $|\widehat{\mathrm{act}}(e_1, [A_2])| \leq |A_2|$, and $|\widehat{\mathrm{act}}([A_1], [A_2])| = |\widehat{\mathrm{act}}([A_1], e_2)| \times |\widehat{\mathrm{act}}(e_1, [A_2])| \leq |A_1| \times |A_2|$. The equality holds when the actions are faithful. Thanks to the monoid structure and the product structure, we can reduce the number of endofunctions that we need to deal with from $|A_1| \times |A_2|$ to at most $|A_1| + |A_2|$. We can further reduce the number if $A_1$ or $A_2$ has a smaller *generator*. For example, although the monoid $(\mathbb{N}, +)$ of natural numbers under addition has infinite elements, it can be generated from a singleton $\{1\}$. In this case, we can focus on a single endofunction that increases the value by a unit, and all other endofunctions are compositions of this special endofunction.

## 4.2 EQUIVARIANCE REQUIREMENT

Then, consider a function $f : X \to Y$ that extracts only necessary information and preserves the algebraic structure of interest. We require it to be equivariant to two actions $\mathrm{act}_X$ and $\mathrm{act}_Y$. Recall that we can consider endofunctions only of the form $\widehat{\mathrm{act}}(a_1, e_2)$ and $\widehat{\mathrm{act}}(e_1, a_2)$. Based on Eq. (1) (*action commutes with transformation*), we can derive the algebraic requirement shown in the following commutative diagram:

$$
\begin{array}{ccccc}
x_{y_1, y_2} & \xrightarrow{f} & (y_1, y_2) & \xrightarrow{p_1} & y_1 \\
\widehat{\mathrm{act}}_X(a_1, e_2) \downarrow & & \widehat{\mathrm{act}}_Y(a_1, e_2) \downarrow \; \searrow^{p_2} & & \downarrow \widehat{\mathrm{act}}_{Y_1}(a_1) \\
& & & y_2 \quad \mathrm{id}_{Y_2} & \\
x_{y_1', y_2} & \xrightarrow{f} & (y_1', y_2) & \xrightarrow{p_1} & y_1'
\end{array}
\tag{6}
$$

With the projections $p_1$ and $p_2$, we can see that this requirement results in the following four conditions: (a) $f_1 = p_1 \circ f$ is equivariant to $\widehat{\mathrm{act}}_X(-, e_2)$ and $\widehat{\mathrm{act}}_{Y_1}$; (b) $f_1$ is invariant to $\widehat{\mathrm{act}}_X(e_1, -)$; and dually, (c) $f_2 = p_2 \circ f$ is equivariant to $\widehat{\mathrm{act}}_X(e_1, -)$ and $\widehat{\mathrm{act}}_{Y_2}$; (d) $f_2$ is invariant to $\widehat{\mathrm{act}}_X(-, e_2)$. The symbol $-$ is a placeholder, into which arguments can be inserted.

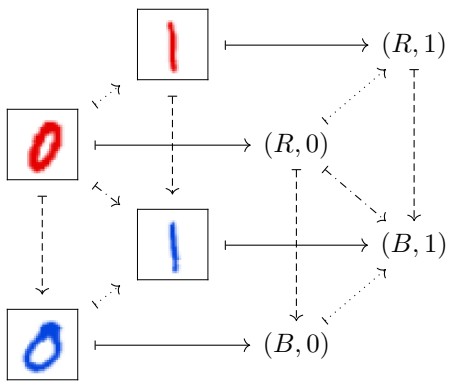

**Notation**:

- component 1    $\vdash\dashrightarrow$   $\widehat{\mathrm{act}}(a_1, e_2)$
- component 2    $\vdash\cdots\rightarrow$   $\widehat{\mathrm{act}}(e_1, a_2)$
- augmentation   $\vdash\rightarrow$   $\widehat{\mathrm{act}}(a_1, a_2)$
- prediction     $\longmapsto$   $f : X \to Y_1 \times Y_2$

**Training**:

- Select suitable data pairs and learn component augmentations separately (Eq. (7));
- Regularize augmentations (Eqs. (8) and (9)), simultaneously or alternatively;
- Train a prediction model (Eq. (10)).

Figure 2: **Equivariant Disentangled Transformation** (EDT). *All diagrams commute.*

### 4.3 ALGORITHM

Finally, we present a method directly derived from the algebraic requirements of the transformation, referred to as *equivariant disentangled transformation* (EDT) and illustrated in Fig. 2. Since the formulation above naturally generalizes to the case of multiple factors $Y = Y_1 \times \cdots \times Y_n$, we present the method in the general form.

**Architecture**   Since the output space $Y$ and the selected endofunctions on it are manually designed, the action $\mathrm{act}_Y$ on $Y$ is known and fixed. However, the action $\mathrm{act}_X$ on $X$ is usually not available. So our first goal is to learn a set of endofunctions $\alpha_i^j : X \to X$ representing $\widehat{\mathrm{act}}_X(e_1, \ldots, a_i^j, \ldots, e_n)$ indexed by $a_i^j \in A_i$, $i = 1, \ldots, n$. These endofunctions can be considered as *learned augmentations* of data that only modify a single factor while keeping other factors fixed. Second, we need to approximate the equivariant function $f$ using a trainable function $\phi : X \to Y$. Due to the property of product, any function to a product arises from component functions $\phi_i : X \to Y_i$, $i = 1, \ldots, n$. Therefore, we can train a model for each component and make these models satisfy the algebraic requirements specified bellow.

**Data selection and augmentation**   To train an augmentation $\alpha_i^j$, we need to collect pairs of instances $x$ and $x'$ such that $\mathrm{act}_X((e_1, \ldots, a_i^j, \ldots, e_n), x) = x'$, in other words, pairs of the form $x_{y_1, \ldots, y_i, \ldots, y_n}$ and $x_{y_1, \ldots, y_i', \ldots, y_n}$, where $\mathrm{act}_{Y_i}(a_i^j, y_i) = y_i'$. Then, denoting the set of all measures on $X$ by $PX$, we can learn the augmentations by minimizing a statistical distance $d : PX \times PX \to \mathbb{R}_{\geq 0}$:

$$\ell_0(\alpha_i^j) = d(\alpha_i^j(x), x'). \tag{7}$$

With a slight abuse of notation, here $x$ and $x'$ also represent the empirical distribution. Choices of the statistical distance $d$ include the expected pairwise distance (Kingma & Welling, 2014), maximum mean discrepancy (Li et al., 2015; Dziugaite et al., 2015; Muandet et al., 2017), Jensen–Shannon divergence (Goodfellow et al., 2014), and Wasserstein metric (Arjovsky et al., 2017; Gulrajani et al., 2017; Miyato et al., 2018).

*Remark* 3 (Cycle consistency). It is possible to use all pairs of the form $x_{y_1, \ldots, y_i, \ldots, y_n}$ and $x_{y_1', \ldots, y_i', \ldots, y_n'}$, i.e., pairs of instances whose $i$-th labels correspond to the action, but other labels could be different. For example, if $A_i$ is a group, we can simultaneously train two models that are the inverse of each other with a cycle consistency constraint (Zhu et al., 2017; Goel et al., 2021). With this constraint, the learned augmentation is likely an approximation of $\widehat{\mathrm{act}}_X(e_1, \ldots, a_i^j, \ldots, e_n)$. However, it is still possible to obtain approximations of $\widehat{\mathrm{act}}_X(q_1, \ldots, a_i^j, \ldots, q_n)$ and its inverse where $q_1, \ldots, q_n$ are not necessarily the identity elements. This happens especially when there are more than two factors and not all combinations are available, which is demonstrated in Section 5.

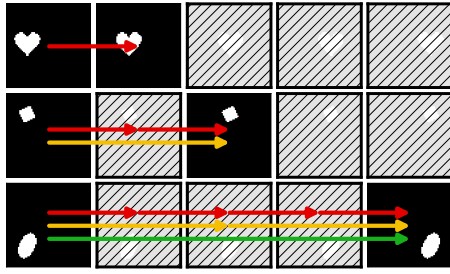

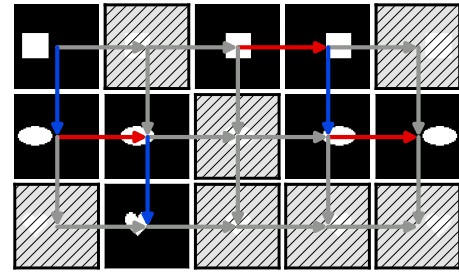

(a) *Compositionality* of multi-scale augmentations

(b) *Commutativity* of two disentangled augmentations

Figure 3: Regularizing compositionality and commutativity (and other algebraic structures) of augmentations is a way to introduce inductive biases and exploit the relationships between training examples, which is useful especially when the combinations of factors are scarce in the training data.

The rich algebraic structure yields various constraints, which can be used as regularization for augmentations. Next, we present three regularization techniques derived from the basic product monoid structure. Note that we can introduce more constraints if we choose a richer algebra.

**Regularization 1 (Compositionality of augmentations)**   According to Eq. (3), if $\alpha_i^{j \cdot k}$ is the approximated action of $a_i^j \cdot_i a_i^k$, we can simply define it as $\alpha_i^{j \cdot k} = \alpha_i^j \circ \alpha_i^k$. If we need to approximate it directly, the algebraic requirement leads to the following regularization:

$$\ell_1(\alpha_i^j, \alpha_i^k, \alpha_i^{j \cdot k}) = d(\alpha_i^j(\alpha_i^k(x)), \alpha_i^{j \cdot k}(x)). \tag{8}$$

A special case is when we know the composition is the identity function $\alpha_i^{j \cdot k} = \mathrm{id}_X$, i.e., $a_i^j$ is the inverse of $a_i^k$. This regularization is then equivalent to the "cycle consistency loss" in the CycleGAN model (Zhu et al., 2017) or the "isomorphism loss" in the GroupifiedVAE model (Yang et al., 2022). Another example is for modifying instances with real-valued targets. We could use multi-scale augmentations (e.g., $\alpha^1$ increases the value by 1 unit and $\alpha^5$ increases the value by 5 units) to reduce the cumulative error and gradient computation, and this regularization ensures that these augmentations are consistent with each other (e.g., $(\alpha^1)^5 \approx \alpha^5$).

**Regularization 2 (Commutativity of augmentations)**   According to the diagram in Eq. (5), we can derive the following regularization, which means that the order of augmentations for different factors should not matter:

$$\ell_2(\alpha_i^k, \alpha_j^l) = d(\alpha_j^l(\alpha_i^k(x)), \alpha_i^k(\alpha_j^l(x))). \tag{9}$$

This can be interpreted as a commutativity requirement: the augmentations are grouped by the factors they modify, and augmentations from different groups should commute, but augmentations within the same group are usually not commutative. Again, we point out that this is only based on the product monoid structure and is nothing group-specific.

In Fig. 4, we illustrate a concrete example of compositionality and commutativity regularization based on the dSprites dataset (Matthey et al., 2017). The movement of position can be modeled via the additive monoid of natural numbers; while the change of shape can be formulated by a permutation/cyclic group. Suitable training example pairs can be used for learning augmentations directly ($\ell_0$), but such pairs may be limited. Algebraic regularization terms (e.g., $\ell_1$ and $\ell_2$) introduce inductive biases so that more relationships between training examples can be used as supervision.

**Regularization 3 (Equivariance of transformation)**   According to the diagram in Eq. (6), we can derive the following equivariance and invariance regularization:

$$\ell_3(\alpha_i^j, \phi_k) = \begin{cases} d(\phi_i(\alpha_i^j(x)), \mathrm{act}_{Y_i}(a_i^j, \phi_i(x))) & i = k, \\ d(\phi_k(\alpha_i^j(x)), \phi_k(x)) & i \neq k. \end{cases} \tag{10}$$

It is a good strategy to learn the augmentations first and then use them to improve the transformation (Goel et al., 2021). However, we can see from this regularization that if the transformation is well trained, it can be used for improving the augmentations too.

Table 1: The classification accuracy (%, "mean (standard deviation)" of 5 trials) on the colored MNIST data. For each setting (column), the method with the highest mean accuracy and those methods that are not statistically significantly different from the best one (via one-tailed t-tests with a significance level of 0.05), if any, are highlighted in boldface.

| (train/test) | AXIS (14/36) | STEP (15/35) | RAND-0.5 (25/25) | RAND-0.7 (35/15) | RAND-0.9 (45/5) |
|---|---|---|---|---|---|
| ERM | 56.74(12.40) | 47.09(8.29) | 91.13(3.70) | 97.18(0.98) | **98.61(0.29)** |
| IRM | 55.40(7.23) | 39.54(7.78) | 87.61(5.37) | 96.89(2.44) | **98.20(0.57)** |
| CORAL | 72.47(17.33) | 49.72(10.53) | 83.48(5.83) | 94.61(3.62) | **98.21(0.75)** |
| DANN | 82.33(12.76) | 45.02(3.79) | 91.76(2.67) | 97.99(0.32) | **98.54(0.20)** |
| Fish | 69.06(14.50) | 45.18(3.36) | 79.93(5.12) | 96.29(1.33) | **98.13(0.43)** |
| Mixup | 63.59(11.98) | 36.30(4.42) | 92.56(1.81) | **97.62(0.99)** | **98.20(0.65)** |
| MixStyle | **97.10(1.36)** | 95.73(1.83) | **95.25(1.83)** | **97.57(1.11)** | **98.13(0.76)** |
| EDT | **97.58(0.17)** | **98.13(0.15)** | **96.70(1.36)** | **98.55(0.13)** | **98.21(0.42)** |

## 5 EXPERIMENTS

As a proof of concept, we conduct experiments to support the following claims:

- Learning data augmentation is a promising approach for the combination shift problem.
- Cycle consistency may be insufficient, and additional constraints need to be considered.
- We should regularize the data augmentations so that they satisfy the algebraic requirements.

### 5.1 COMBINATION SHIFT

First, we experimentally demonstrate the insufficiency of the invariance-based approach and the potential of the augmentation-based approach for the combination shift problem.

**Data** We colored the grayscale images from the MNIST dataset (LeCun et al., 1998) with 5 colors to create a semi-synthetic setting. Therefore, there are 5 domains (colors) and 10 classes (digits). We tested the methods in the most extreme case where the combinations of domains and classes of the training and test sets are disjoint. We selected five types of combinations as the training set: **AXIS**: all red digits and zeros of all colors; **STEP**: three digits for each color (shown in Fig. 6 in Appendix E); **RAND**-0.5/-0.7/-0.9: combinations randomly selected with a fixed ratio.

**Method** In addition to an **ERM** baseline, we evaluated four invariance-based methods: **IRM** (Arjovsky et al., 2019), **CORAL** (Sun & Saenko, 2016), **DANN** (Ganin et al., 2016), and **Fish** (Shi et al., 2022); and two augmentation-based methods: **Mixup** (Zhang et al., 2018) and **MixStyle** (Zhou et al., 2021). Model architectures and hyperparameters are given in Appendix E.

**Results** We can see from Table 1 that the ERM baseline and invariance-based methods perform poorly if only limited combinations of domains and classes are observable. The high variance indicates that the learned representation may still depend on the domains. As more combinations become observable in training, the differences in performance of all methods become less statistically significant. On the other hand, the augmentation-based methods usually provide higher performance improvements, although the mixup method may deteriorate performance depending on the setting. MixStyle performs consistently well, partially because it is specifically designed for image styles and thus lends itself well to this setting. With the algebraic constraints, EDT may capture the underlying distribution better and offer larger improvements.

### 5.2 DATA AUGMENTATION

Next, we discuss potential issues of the augmentation-based method (Goel et al., 2021) based on CycleGAN (Zhu et al., 2017), which matches the bidirectionally transformed distributions and regularizes the composition to be the identity functions. There are two major issues of this approach. Firstly, it is designed only for two domains (e.g., female and male). Secondly and more importantly,

Table 2: The *misclassification rate* (%) of shape and *mean squared errors* ($\times 100$) of scale, orientation, and positions on the dSprites dataset ("mean (standard deviation)" of 5 trials).

| | Shape | Scale | Orientation | Position X | Position Y |
|---|---|---|---|---|---|
| ERM | 60.84(2.24) | 3.76(0.24) | 13.13(0.72) | 1.97(0.69) | 1.87(0.25) |
| MixStyle | 59.92(2.00) | 6.73(1.07) | 13.04(0.54) | 0.20(0.10) | 0.21(0.06) |
| EDT ($\ell_0, \ell_3$) | 14.36(0.75) | 1.30(0.06) | **2.09(0.07)** | 0.04(0.01) | 0.04(0.01) |
| EDT ($\ell_0, \ell_1, \ell_2, \ell_3$) | **4.55(0.21)** | **0.59(0.01)** | **2.01(0.07)** | **0.02(0.00)** | **0.02(0.00)** |



(a) Without *compositionality regularization*, the error may accumulate after a few compositions.

(b) Without *commutativity regularization*, pairs for learning augmentations may be insufficient.

Figure 4: Randomly selected 5 images (top row) in the dSprites dataset (Matthey et al., 2017) and augmented images (bottom 4 rows) of position (Fig. 4a, left ⤳ right) and shape (Fig. 4b, square ⤳ ellipse ⤳ heart ⤳ square), without (left) and with (right) regularization.

as discussed in Remark 3, when there are more than two factors, cycle consistency alone may not guarantee the identity of non-transformed factors. The comparison on the 3D Shapes dataset (Burgess & Kim, 2018) is shown in Fig. 8 in Appendix E. We can observe that although the floor hue is transformed as desired and the reconstructed images are almost identical to the original ones, other factors such as the object/wall hues are also changed. In contrast, the algebraic requirements of EDT ensure the approximated augmentations are consistent with the desired actions.

### 5.3 ALGEBRAIC REGULARIZATION

Finally, we further compare heuristic and learned data augmentations and demonstrate the usefulness of algebraic regularization. We used the dSprites dataset (Matthey et al., 2017) and considered one factor as target label and the others as domains. Some methods are no longer applicable because of the continuous or even periodic values of factors and the multiplicatively increasing number of combinations. In Table 2, we can see that MixStyle provides no significant performance gain in this setting because the heuristic augmentation does not match the underlying mechanism anymore (See also Fig. 10 in Appendix E). In Fig. 4, we provide the results of an ablation study of the compositionality ($\ell_1$) and commutativity ($\ell_2$) regularization, showing that these regularization terms can reduce errors accumulated by compositions of augmentations and increase the number of supervision signals for learning augmentations, as illustrated in Fig. 3.

## 6 CONCLUSION

Unlike the usual goal of generalizing to an unseen domain, we formulated the problem of combination shift as learning the knowledge of each factor (domains and labels) and generalizing to unseen combinations of factors, which makes deployment more feasible but training more challenging. We found that invariance-based methods may not work well in this setting, but augmentation-based methods usually excel. To formally analyze data augmentations and provide a guideline on augmentation design, we presented an algebraic formulation of the problem, which also leads to a refined definition of disentanglement. We demonstrated the usefulness of constraints derived from algebraic requirements, discussed potential issues of the existing augmentation method based on cycle consistency, and showed the importance of algebraic regularization. We then pointed out several promising research directions, such as incorporating algebra homomorphism and multi-sorted algebra to discuss a wider range of data augmentation operations. We hope that our algebraic formulation can be used to derive practical algorithms in applications and inspire further studies in this direction.

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

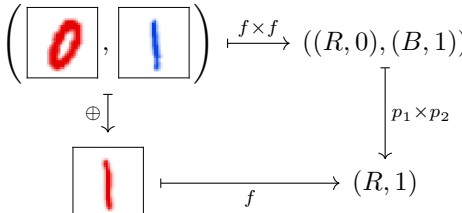

Figure 5: A homomorphism preserving binary operations $\oplus$ and $p_1 \times p_2$.

## A   LIMITATIONS AND FUTURE WORK

In this section, we discuss the limitations of this work and potential future work directions.

### A.1   ALGEBRA HOMOMORPHISM

In this work, we only formulated data augmentations of the endofunction form $\alpha : X \to X$, i.e., modifications of only one input. However, there are other operations that do not fall into this form. We suggest using algebra homomorphisms to capture their relations. Here we give three examples:

**Component combination**   If the instance can be divided into multiple components, then we can recombine the components from multiple instances to generate a new instance: $\alpha : X^n \to X$. This is especially useful when there are many factors and the combinations in the training set are sparse.

**Style transfer**   Another example is when we cannot divide the instances but can combine their characteristics, such as style transfer (Gatys et al., 2016). An example is given in Fig. 5, where $\oplus : X \times X \to X$ is the binary operation that takes the "style" of the first image and the "content" of the second image, and $p_1 \times p_2 : Y \times Y \to Y$ is the corresponding operation in the label space $Y$. Then, we need to ensure that this binary operation is compatible with other augmentations. For example, if the object in the content image changes, the object in the generated image should change accordingly; while the generated image should not change regardless of the object in the style image.

**Crowd counting**   Counting the number of objects or people in an image is an example where we can exploit the structure of natural numbers $\mathbb{N}$. In addition to the monotone function requirement induced by the total order of natural numbers $\mathbb{N}$ (Liu et al., 2018), the free monoid structure $(\mathbb{N}, +)$ may induce other useful constraints. For example, the count of two parts should be the sum of the counts of each part. This requirement can be formulated as an algebra homomorphism.

### A.2   STATISTICS AND APPROXIMATION

Similarly to previous work (Higgins et al., 2018), we focused more on the algebraic aspect. We admit that there is still a gap between formulation and practice, because algebra only describes exact equality ($=$), but sometimes we are more interested in approximate equality ($\approx$). It would be useful to define concepts such as *commutativity over a metric space*, so that we can analyze errors and introduce statistical tools, to get the best of both worlds.

### A.3   STATE AND MULTI-SORTED ALGEBRA

Another issue is that we only considered endofunctions $X \to X$ so all data augmentations are applicable to all instances in a "stateless" way, which may not hold true in more complex situations. As a future work, we may consider general functions $X_i \to X_j$ and define which functions are composable and which are not.

Also, it could be useful to discuss operations on multiple sets based on multi-sorted algebra, such as graphs (de Haan et al., 2020).

## B   A BRIEF REVIEW OF ALGEBRA

In this section, we review the algebraic concepts used in this work. We refer the readers to Dummit & Foote (1991) (abstract algebra), Bergman (2015) (universal algebra), and Awodey (2010) (category theory) for further readings.

### B.1   ALGEBRA

**Definition 2** (Algebra).  A (single-sorted) **algebra** consists of

- a *set* $A$, called the underlying set of the algebra,

- a collection of *operations* $\{f^i : A^{n_i} \to A\}_{i \in I}$, and

- a collection of universally quantified equational *axioms* that those operations satisfy.

For example, elementary algebra is the study of the set of numbers with arithmetic operations such as addition, subtraction, multiplication, division, and exponentiation. Linear algebra is the study of the set of vectors with operations of vector addition and scalar multiplication.

Some algebras with only one binary operation are listed below.

**Definition 3** (Magma).  A **magma** is a set $A$ equipped with a binary operation $\cdot : A \times A \to A$.

**Definition 4** (Semigroup).  A **semigroup** is a magma $(S, \cdot)$ whose binary operation is *associative*:

$$\forall s_1, s_2, s_3 \in S, (s_1 \cdot s_2) \cdot s_3 = s_1 \cdot (s_2 \cdot s_3). \tag{11}$$

**Definition 5** (Monoid).  A **monoid** is a semigroup $(M, \cdot)$ that has an *identity element* $e \in M$ (a nullary operation $e : 1 \to M$):

$$\forall m \in M, e \cdot m = m \cdot e = m. \tag{12}$$

**Definition 6** (Group).  A **group** is a monoid $(G, \cdot, e)$, and every element has an *inverse* (a unary operation $(-)^{-1} : G \to G$):

$$\forall g \in G, g \cdot g^{-1} = g^{-1} \cdot g = e. \tag{13}$$

**Definition 7** (Abelian group).  An **Abelian group** is a group $(G, \cdot, e, (-)^{-1})$ whose binary operation is *commutative*:

$$\forall g_1, g_2 \in G, g_1 \cdot g_2 = g_2 \cdot g_1. \tag{14}$$

### B.2   HOMOMORPHISM

**Definition 8** (Homomorphism).  A **homomorphism** between two algebras $(A, \{f_A^i\}_{i \in I})$ and $(B, \{f_B^i\}_{i \in I})$ of the same type is a function between the underlying sets $h : A \to B$ such that

$$\forall a_1, \ldots, a_{n_i} \in A, h(f_A^i(a_1, \ldots, a_{n_i})) = f_B^i(h(a_1), \ldots, h(a_{n_i})) \tag{15}$$

holds for all corresponding operations $f_A^i : A^{n_i} \to A$ and $f_B^i : B^{n_i} \to B$.

In other words, the following diagram commutes for all $i \in I$:

$$
\begin{array}{ccc}
A^{n_i} & \xrightarrow{h^{n_i}} & B^{n_i} \\
{\scriptstyle f_A^i}\downarrow & & \downarrow{\scriptstyle f_B^i} \\
A & \xrightarrow{\;\;h\;\;} & B
\end{array}
\tag{16}
$$

An invertible homomorphism is called an isomorphism. For example, $\exp$ and $\log$ functions form a pair of isomorphisms between $(\mathbb{R}, +)$ and $(\mathbb{R}^+, \times)$ because $\exp(x + y) = \exp(x) \times \exp(y)$ and $\log(x \times y) = \log(x) + \log(y)$.

## B.3 EXPONENTIAL

**Definition 9** (Exponential). Given sets $A$ and $B$, the **function set** $B^A$ is the set of all functions from $A$ to $B$. Given a set $A$ and a function set $B^A$, there exists an **evaluation map** $\epsilon : B^A \times A \to B$ that sends a function $f : A \to B$ and a value $a \in A$ to the evaluation $\epsilon(f, a) = f(a) \in B$.

**Definition 10** (Exponential transpose). For a binary function $f : A \times B \to C$, its **exponential transpose** (also known as **currying**) is a function $\widehat{f} : A \to C^B$ such that

$$\forall a \in A, \forall b \in B, f(a, b) = \widehat{f}(a)(b). \tag{17}$$

## B.4 ACTION

**Definition 11** (Action). A (left) **action** of a set $A$ on a set $X$ is a binary function $\mathrm{act} : A \times X \to X$.

**Definition 12** (Representation). A **representation** of a set $A$ on a set $X$ is a function $\widehat{\mathrm{act}} : A \to X^X$.

**Definition 13** (Algebra preservation). A representation $\widehat{\mathrm{act}} : A \to X^X$ preserves an algebra over $A$ if it is a homomorphism from $A$ to $X^X$.

A magma/semigroup action preserves composition (a binary operation):

$$\forall a_1, a_2 \in A, \forall x \in X, \mathrm{act}(a_1 \cdot a_2, x) = \mathrm{act}(a_1, \mathrm{act}(a_2, x)). \tag{18}$$

$$\begin{array}{ccc} A \times A \times X & \xrightarrow{\mathrm{id}_A \times \mathrm{act}} & A \times X \\ {\scriptstyle \cdot \times \mathrm{id}_X} \downarrow & & \downarrow {\scriptstyle \mathrm{act}} \\ A \times X & \xrightarrow{\mathrm{act}} & X \end{array} \tag{19}$$

Or equivalently,

$$\forall a_1, a_2 \in A, \widehat{\mathrm{act}}(a_1 \cdot a_2) = \widehat{\mathrm{act}}(a_1) \circ \widehat{\mathrm{act}}(a_2). \tag{20}$$

$$\begin{array}{ccc} A \times A & \xrightarrow{\widehat{\mathrm{act}} \times \widehat{\mathrm{act}}} & X^X \times X^X \\ {\scriptstyle \cdot} \downarrow & & \downarrow {\scriptstyle \circ} \\ A & \xrightarrow{\widehat{\mathrm{act}}} & X^X \end{array} \tag{21}$$

A monoid action preserves identity (a nullary operation):

$$\forall x \in X, \mathrm{act}(e, x) = x. \tag{22}$$

$$\widehat{\mathrm{act}}(e) = \mathrm{id}_X . \tag{23}$$

$$\begin{array}{ccc} 1 & \longrightarrow & 1 \\ {\scriptstyle e} \downarrow & & \downarrow {\scriptstyle \mathrm{id}_X} \\ A & \xrightarrow{\widehat{\mathrm{act}}} & X^X \end{array} \tag{24}$$

A group action preserves inverse (a unary operation):

$$\forall a \in A, \forall x \in X, \mathrm{act}(a^{-1}, \mathrm{act}(a, x)) = x. \tag{25}$$

$$\forall a \in A, \widehat{\mathrm{act}}(a^{-1}) = \widehat{\mathrm{act}}(a)^{-1}. \tag{26}$$

$$\begin{array}{ccc} A & \xrightarrow{\widehat{\mathrm{act}}} & X^X \\ {\scriptstyle (-)^{-1}} \downarrow & & \downarrow {\scriptstyle (-)^{-1}} \\ A & \xrightarrow{\widehat{\mathrm{act}}} & X^X \end{array} \tag{27}$$

## B.5 EQUIVARIANCE

**Definition 14** (Equivariance). A function $f : X \to Y$ is **equivariant** to two actions $\mathrm{act}_X : A \times X \to X$ and $\mathrm{act}_Y : A \times Y \to Y$ if

$$\forall a \in A, \forall x \in X, f(\mathrm{act}_X(a, x)) = \mathrm{act}_Y(a, f(x)). \tag{28}$$

$$
\begin{array}{ccc}
A \times X & \xrightarrow{\ \mathrm{id}_A \times f\ } & A \times Y \\
{\scriptstyle \mathrm{act}_X} \downarrow & & \downarrow {\scriptstyle \mathrm{act}_Y} \\
X & \xrightarrow[\ \ f\ \ ]{} & Y
\end{array}
\tag{29}
$$

Or equivalently,

$$\forall a \in A, f \circ \widehat{\mathrm{act}}_X(a) = \widehat{\mathrm{act}}_Y(a) \circ f. \tag{30}$$

$$
\begin{array}{ccc}
X & \xrightarrow{\ \ f\ \ } & Y \\
{\scriptstyle \widehat{\mathrm{act}}_X(a)} \downarrow & & \downarrow {\scriptstyle \widehat{\mathrm{act}}_Y(a)} \\
X & \xrightarrow[\ \ f\ \ ]{} & Y
\end{array}
\tag{31}
$$

commutes for all $a \in A$. This justifies that an equivariant map is a homomorphism between two algebras whose operations are all unary and indexed by elements in the set $A$.

## B.6 PRODUCT

**Definition 15** (Product). A **product** $A \times B$ of two objects $A$ and $B$ and the corresponding **projections** $p_1 : A \times B \to A$ and $p_2 : A \times B \to B$ satisfy that for any object $C$ and morphisms $f_1 : C \to A$ and $f_2 : C \to B$, there is a unique morphism $f : C \to A \times B$, such that $f_1 = p_1 \circ f$ and $f_2 = p_2 \circ f$, as indicated in

$$
\begin{array}{ccc}
 & C & \\
{\scriptstyle f_1} \swarrow & \downarrow {\scriptstyle f} & \searrow {\scriptstyle f_2} \\
A \xleftarrow{\ p_1\ } & A \times B & \xrightarrow{\ p_2\ } B
\end{array}
\tag{32}
$$

Consider two morphisms $f : C \to A$ and $g : D \to B$. Based on the universal property of $A \times B$, there exists a unique morphism $f \times g : C \times D \to A \times B$ such that the following diagram commutes:

$$
\begin{array}{ccccc}
C & \xleftarrow{\ p_1\ } & C \times D & \xrightarrow{\ p_2\ } & D \\
{\scriptstyle f} \downarrow & & \downarrow {\scriptstyle f \times g} & & \downarrow {\scriptstyle g} \\
A & \xleftarrow[\ p_1\ ]{} & A \times B & \xrightarrow[\ p_2\ ]{} & B
\end{array}
\tag{33}
$$

For example, let both $C$ and $D$ be $Y_1 \times Y_2$, $f = p_1$, and $g = p_2$. Then, the following diagram represents "recombination of components":

$$
\begin{array}{ccccc}
Y_1 \times Y_2 & \xleftarrow{\ p_1\ } & (Y_1 \times Y_2) \times (Y_1 \times Y_2) & \xrightarrow{\ p_2\ } & Y_1 \times Y_2 \\
{\scriptstyle p_1} \downarrow & & \downarrow {\scriptstyle p_1 \times p_2} & & \downarrow {\scriptstyle p_2} \\
Y_1 & \xleftarrow[\ p_1\ ]{} & Y_1 \times Y_2 & \xrightarrow[\ p_2\ ]{} & Y_2
\end{array}
\tag{34}
$$

# C  ALGEBRA IN SUPERVISED LEARNING

In this section, we look ahead to the application of algebraic theory to supervised learning.

## C.1  SUPERVISED LEARNING

Let $X$ be the set of inputs and $Y$ the set of outputs. In supervised learning, we want to find a function $f : X \to Y$ that satisfies some properties. Generally, this is achieved by collecting a set of pairs $\{(x_i, y_i) \in X \times Y\}_{i \in I}$ as training examples and defining a measure of "goodness" of functions. For example, for a pair $(x_i, y_i)$, we expect $f$ to map $x_i$ to $y_i$.

Let us consider this procedure from an algebraic perspective.

**Nullary operation**  First, we point out that identifying an element $x$ from a set $X$ can be considered as a nullary operation $x : 1 \to X$, and evaluating a function $f : X \to Y$ at an element $x$ is simply function composition $f \circ x : 1 \to Y$. Then, requiring

$$f(x) = y \tag{35}$$

is equivalent to say that $f$ should be an *algebra homomorphism*:

$$
\begin{array}{ccc}
1 & \longrightarrow & 1 \\
{\scriptstyle x}\downarrow & & \downarrow{\scriptstyle y} \\
X & \xrightarrow{\ f\ } & Y
\end{array}
\tag{36}
$$

Therefore, a function that can predict all training examples perfectly is simply a homomorphism from algebra $(X, \{x_i : 1 \to X\}_{i \in I})$ to algebra $(Y, \{y_i : 1 \to Y\}_{i \in I})$ where all operations are nullary.

This perspective frames direct supervision as an algebraic requirement. However, it is still not practically useful, because the training examples are usually finite and cannot enumerate the set of inputs, but we need machine learning only when the inputs in a test environment are not exactly the same as the inputs for training. Two things are missing: first, we need an assumption to relate training and test data; second, we need not only "yes or no" but also "how much". As discussed in Appendix A, pure algebra only deals with exact equality, so integrating algebra and statistical learning is an important research direction.

**Unary operation**  Many works introducing algebraic theory, especially group theory, into machine learning, including this work, have focused on unary operations and their relations. A unary operation or an endofunction $\alpha_X : X \to X$ transforms a set of states to itself. A homomorphism between $(X, \alpha_X)$ and $(Y, \alpha_Y)$ just relates these unary operations:

$$
\begin{array}{ccc}
X & \xrightarrow{\ f\ } & Y \\
{\scriptstyle \alpha_X}\downarrow & & \downarrow{\scriptstyle \alpha_Y} \\
X & \xrightarrow{\ f\ } & Y
\end{array}
\tag{37}
$$

Usually, there are multiple unary operations, which themselves form an algebra. Magma/semigroup describes composition, monoid describes identity, and group describes invertibility. An invertible unary operation/endofunction is also called a symmetry. The structure of these unary operations can be described by an action preserving the algebraic structure, which was extensively used in this work.

**Binary and $n$-ary operations**  As also covered in Appendix A, not all operations are unary operations. It would be useful to include $n$-ary operations and their relations as algebraic requirements for $f$:

$$
\begin{array}{ccc}
X^n & \xrightarrow{\ f^n\ } & Y^n \\
{\scriptstyle \alpha_X}\downarrow & & \downarrow{\scriptstyle \alpha_Y} \\
X & \xrightarrow{\ f\ } & Y
\end{array}
\tag{38}
$$

Specifically, operad theory could be useful for analyzing a collection of finitary operations obeying equational axioms.

Moreover, future research could continue to explore $n$-ary functions from an algebraic perspective. For example, $f : X \to Y$ and $g : A \to B$ may relate two binary functions $\alpha_X : X \times X \to A$ and $\alpha_Y : Y \times Y \to B$ in the following sense:

$$
\begin{array}{ccc}
X \times X & \xrightarrow{f \times f} & Y \times Y \\
{\scriptstyle \alpha_X} \downarrow & & \downarrow {\scriptstyle \alpha_Y} \\
A & \xrightarrow{\quad g \quad} & B
\end{array}
\tag{39}
$$

which could be used for formulating *relation-preserving functions*, such as equality (learning from similarity) and order (learning to rank), or *metrics*, such as isometry, contraction, and Lipschitz continuous function.

## C.2 BINARY CLASSIFICATION

Now, let us consider a concrete example, binary classification. Let $\bar{n}$ be a set whose cardinality is $n$, $\bar{1}$ a *singleton* (a set of a single element), $+$ the *disjoint union* of sets (union of labeled/indexed elements), $\cong$ the isomorphism between two sets (a bijective function). In binary classification, $Y$ is simply a set of two elements $\bar{2} \cong \bar{1} + \bar{1}$.

In other words, we only have a space with the concept of *sameness* or *equality* and no other operations. The learning process is to find a function $f : X \to \bar{2}$, which decomposes into a pair of functions $f = f_1 + f_2$, where $f_i : X_i \to \bar{1} (i = 1, 2)$. This results in a decomposition of $X$ into two sets $X \cong X_1 + X_2$, i.e., classification of elements in $X$.

Let us examine the unary operations (endofunctions) on $\bar{2}$. There are in total four endofunctions on $\bar{2}$, which forms a monoid. There are only two invertible ones: the identity and the one that swaps two elements, which constitute a representation of the *symmetric group* $S_2$ on $\bar{2}$.

## C.3 REGRESSION

To formulate regression, we usually let $Y$ be the set of real numbers $\mathbb{R}$. However, from an algebraic perspective, many operations of real numbers are not needed in the learning process. For example, we rarely consider the product or ratio of two target values. On the other hand, the order, scale, and zero point are of our central interest. Thus, if there exist a *minimal value* and a *unit interval* of targets, we can isomorphically transform the target and let $Y$ be the set of natural numbers $\mathbb{N}$. If we cannot determine a minimal value but we are still able to quantize the target values, we can take a step further and consider the algebra of integers $\mathbb{Z}$ and the negation operation.

There are two important operations of natural numbers: $0 : 1 \to \mathbb{N}$ as a nullary operation that identifies the number zero and the *successor function* $S : \mathbb{N} \to \mathbb{N}$ as a unary operation that maps a number $n$ to the next number $S(n)$.

Let $x_n \in X$ be an instance whose label is $n$. If $X$ also has the structure of natural numbers, then there exist an element $x_0$ that has the minimal value and a unary operation $T : X \to X$ that takes an instance as input and outputs another instance whose label is one unit higher. The requirement of $f$ being a homomorphism means that the instance with the minimal value is mapped to $0$, i.e., $f(x_0) = 0$, and the operation $T$ corresponds to the successor function $S$ in the following way:

$$
\begin{array}{ccc}
x_n & \xmapsto{\quad f \quad} & n \\
{\scriptstyle T} \uparrow \downarrow & & \downarrow {\scriptstyle S} \\
x_{S(n)} & \xmapsto{\quad f \quad} & S(n)
\end{array}
\tag{40}
$$

Given the number zero $0$ and the successor function $S$ of natural numbers $\mathbb{N}$, we can define a *commutative monoid* with $0$ as the identity element and a monoid operation $+$ defined recursively: $a + S(b) := S(a + b)$. This is the *free monoid* $(\mathbb{N}, +)$ generated from a *generator* $\{1 := S(0)\}$.

Then, we can consider the case when the free monoid $(\mathbb{N}, +)$ acts on $X$ and $\mathbb{N}$ itself. A function equivariant to free monoid actions is a function $f : X \to \mathbb{N}$ such that the following diagram commutes:

$$
\begin{array}{ccc}
(m, x_n) & \xmapsto{\ \mathrm{id}_\mathbb{N} \times f\ } & (m, n) \\
{\scriptstyle \mathrm{act}_X} \Big\downarrow & & \Big\downarrow {\scriptstyle +} \\
x_{n+m} & \xmapsto[f]{} & n + m
\end{array}
\tag{41}
$$

Note that when $m$ is the generator 1, this diagram can be reduced to Eq. (40).

The crowd counting example in Appendix A can be illustrated in the following diagram:

$$
\begin{array}{ccc}
(x_m, x_n) & \xmapsto{\ f \times f\ } & (m, n) \\
{\scriptstyle \oplus} \Big\downarrow & & \Big\downarrow {\scriptstyle +} \\
x_{m+n} & \xmapsto[f]{} & m + n
\end{array}
\tag{42}
$$

which means that the count of two parts should be the sum of the counts of each part. This requirement is formulated as a homomorphism of binary operations $\oplus : X \times X \to X$ and $+ : \mathbb{N} \times \mathbb{N} \to \mathbb{N}$.

## C.4 DISCUSSION

As discussed in Section 3.1, the equivariance alone may not fully characterizes a learning problem. For example, in binary classification, if we only require the transformation $f : X \to Y$ to be equivariant to actions by the symmetric group $S_2$, then $f$ is only *unique up to permutation*; Similarly, in regression, $f$ is only *unique up to shift* by a natural number or an integer. This may not cause a problem, but we still need some information to determine the optimal solution, for example, the zero point (a nullary operation) in regression.

Similarly to Higgins et al. (2018), we focused on the algebraic aspect of disentanglement. It is worth noting that this formulation is not yet compatible with some definitions of disentanglement based on statistical independence, probability metric, or causal mechanisms (Higgins et al., 2017; Suter et al., 2019; Locatello et al., 2019; Shu et al., 2020; Tokui & Sato, 2022). In statistical learning, we usually want to find a *conditional distribution* $\bar{f} : X \to PY$, where $PY$ denotes all probability measures on $Y$, instead of merely a deterministic transformation $f : X \to Y$. To extend this framework and fully capture the statistical aspect of disentanglement, we need to further incorporate the structure of probability measures, which is left for future work.

# D    LITERATURE REVIEW: DISTRIBUTION SHIFT

In this section, we review related work in distribution shift in a broader sense.

The difference between the training and test data in supervised learning is an important problem and has been studied for years. The *distribution shift* problem (Quiñonero-Candela et al., 2008) refers to the general case where the training and test data are drawn from related but different distributions:

$$p^{\text{train}}(X, Y) \neq p^{\text{test}}(X, Y)$$

The difference can be measured by some distribution divergence (Ben-David et al., 2010; Albuquerque et al., 2019). Distribution shift can be subcategorized by the distribution assumptions:

- *Covariate shift*: $p^{\text{train}}(Y \mid X) = p^{\text{test}}(Y \mid X)$ (Sugiyama et al., 2007; Sugiyama & Kawanabe, 2012)
- *Label shift*: $p^{\text{train}}(X \mid Y) = p^{\text{test}}(X \mid Y)$, e.g., class imbalance (Johnson & Khoshgoftaar, 2019) and long-tailed class distribution (Zhang et al., 2021)
- *Concept shift*: $p^{\text{train}}(X) = p^{\text{test}}(X)$, e.g., noisy labels (Song et al., 2022)

Distribution shift is also closely related to *robust optimization* (Ben-Tal & Nemirovski, 2002) and *fairness* in machine learning (Barocas et al., 2019).

*Domain adaptation/generalization* (Wang et al., 2021a) is a special distribution shift problem (Quiñonero-Candela et al., 2008), implying that the tasks are indexed by a categorical (Blanchard et al., 2011) or continuous (Wang et al., 2020) domain variable. All three types of distribution shift mentioned above may happen when there are multiple domains. To solve this problem, *domain-invariant representation learning* (Ganin et al., 2016; Sun & Saenko, 2016; Arjovsky et al., 2019; Creager et al., 2021; Shi et al., 2022) has been widely used, which aims to extract features invariant to domain change. In this work, we showed the limitations of invariance-based methods in the combination shift problem.

A closely related concept is *disentanglement* (Bengio et al., 2013), which can be defined via statistical independence (Suter et al., 2019; Locatello et al., 2019; Shu et al., 2020; Tokui & Sato, 2022) or product group action (Higgins et al., 2018; Caselles-Dupré et al., 2019; Quessard et al., 2020; Painter et al., 2020; Wang et al., 2021b; Yang et al., 2022). Our work follows the latter direction. We provided a refined definition of disentanglement based on algebra in Definition 1, which can be seen as an extension of Higgins et al. (2018). We also discussed potential directions for further extension in Appendix A, including algebra homomorphism, statistics, non-endofunctions, and multi-sorted algebra.

Various methods have been developed based on the concept of disentanglement. On approach is based on variants of the *variational autoencoder* (VAE) (Kingma & Welling, 2014; Higgins et al., 2017). Another promising approach is based on either heuristic (Zhang et al., 2018; Shorten & Khoshgoftaar, 2019; Chen et al., 2020; Zhou et al., 2021) or learned (Ratner et al., 2017; Volpi et al., 2018; Wang et al., 2021c; Goel et al., 2021) *data augmentation*. Learning data augmentation is the central interest of our work.

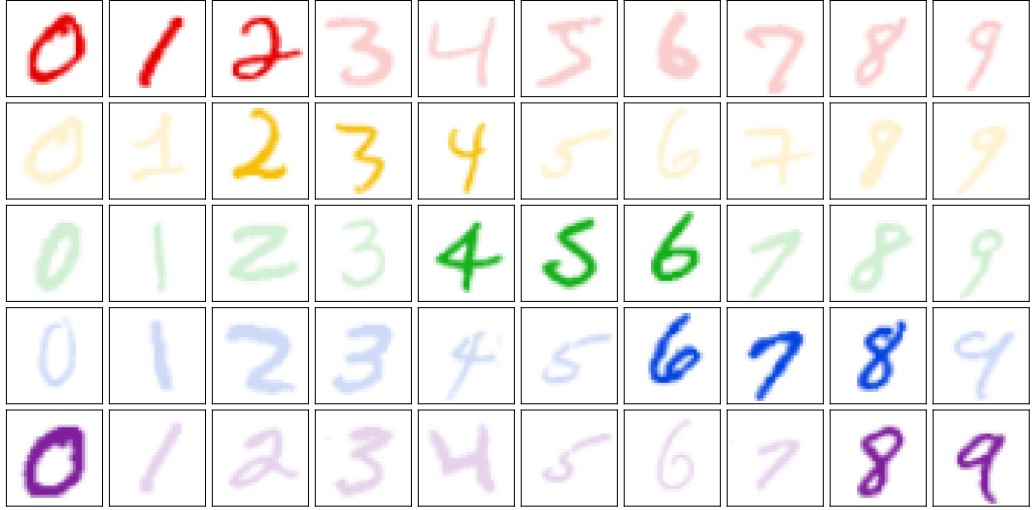

Figure 6: A set of combinations of the colored MNIST data with only $15/50 = 30\%$ data for training. Shaded combinations are used for testing.

## E EXPERIMENTS

### E.1 MNIST

**Data**  The MNIST[1] dataset contains grayscale hand-written digit images of size $28 \times 28$ in 10 classes. The size of the training set is $60\,000$ and the size of the test set is $10\,000$. We only used the images in the training set and colored them with five colors (red, yellow, green, blue, and purple) with equal probabilities. The images were resized to $32 \times 32$ to fit the model. No manual data augmentation was used.

**Data split**  We selected five types of combinations as the training set:

- **AXIS**: all red digits and zeros of all colors
- **STEP**: three digits for each color, shown in Fig. 6
- **RAND**-0.5/-0.7/-0.9: combinations randomly selected with a fixed ratio 0.5, 0.7, or 0.9. All domains and classes were ensured to appear at least once.

The remaining combinations were used as the test set.

**Model**  We used U-Net (Ronneberger et al., 2015) for the image-to-image data augmentations with 3 layers of downscale/upscale modules and a sigmoid as the last layer. We used a convolutional neural network with spectral norm (Miyato et al., 2018) as the discriminator for distribution matching (Goodfellow et al., 2014) between images ($\ell_0$, $\ell_1$, and $\ell_2$). To reduce the number of models, the discriminator was conditioned on the factors via additive embedding. We use the same architecture of the discriminator for the classifier except the dimension of output was set to 10. The learning objective for the classifier ($\ell_3$) is the cross-entropy/negative log-likelihood.

**Optimization**  We used an Adam optimizer (Kingma & Ba, 2015) with batch size of 32, learning rate of $1 \times 10^{-3}$ for the augmentations and $1 \times 10^{-4}$ for the discriminator and the classifier. The model was trained for $10\,000$ iterations.

**Infrastructure**  The experiments were conducted on an NVIDIA Tesla V100 GPU.

---

[1]MNIST (LeCun et al., 1998) http://yann.lecun.com/exdb/mnist/

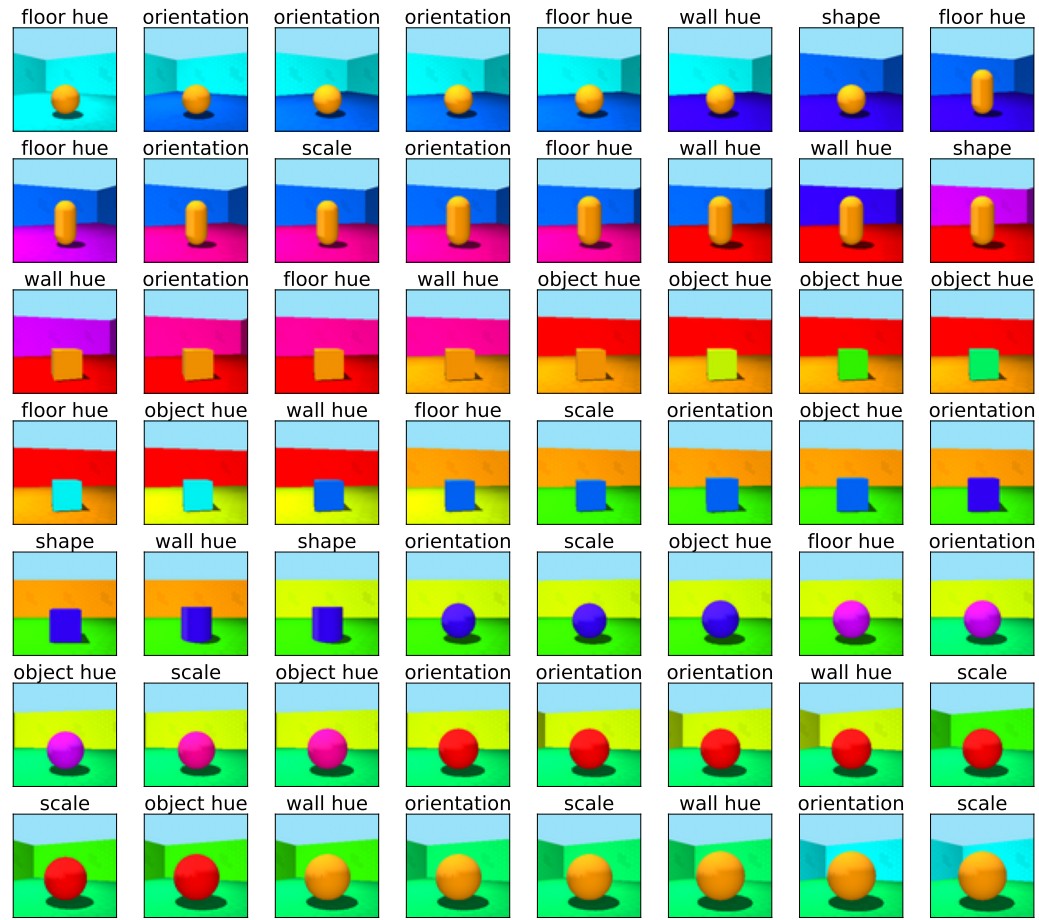

Figure 7: A path of transformations of data (left to right, top to bottom) of the 3D Shapes dataset.

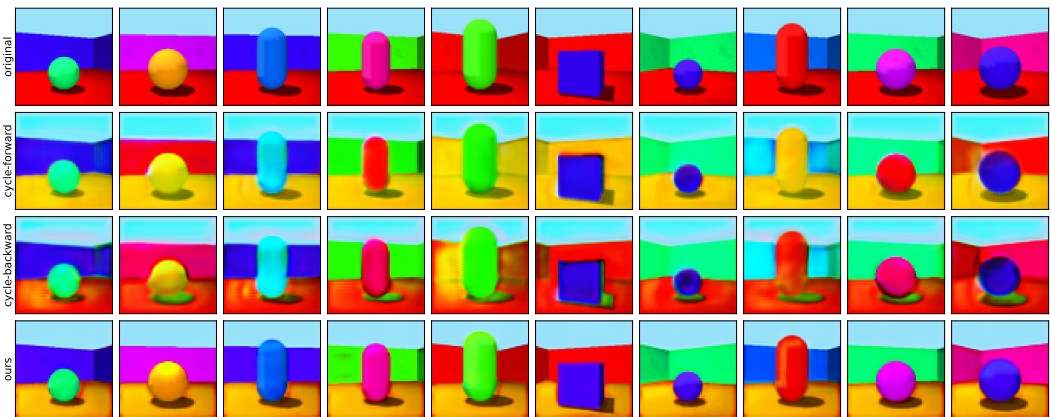

Figure 8: (top row) 10 images from the 3D Shapes dataset (Burgess & Kim, 2018) with red floor; (middle rows) augmented data (red to orange) and reconstructed data (orange to red) transformed by a CycleGAN model (Zhu et al., 2017; Goel et al., 2021); (bottom row) augmented data transformed by EDT, which satisfies the algebraic constraints.

### E.2 3D Shapes

**Data** The 3D Shapes[2] dataset contains images of three-dimensional objects with 6 factors (floor hue, wall hue, object hue, scale, shape, and orientation), whose dimensions are 10, 10, 10, 8, 4, and 15. The size of the dataset is $480\,000$.

**Data selection** Since the goal is to improve generalization using as few combinations as possible, we used a set of properly selected combinations of factors. Concretely, we first randomly select an instance, and then randomly change a factor at a time. An example of a path of transformations is shown in Fig. 7. We used 10 random paths so there are at most 570 training examples (only around $0.1\%$ of all data).

**Model and optimization** Because there is only one image for each combination of factors, there is no need to use distribution matching. We used pixel-wise binary cross-entropy as the learning objective for $\ell_0$, $\ell_1$, and $\ell_2$. Other hyperparameters are the same as those used above.

### E.3 dSprites

**Data** The dSprites[3] dataset contains images of 2D shapes generated from 6 ground truth independent latent factors: color, shape, scale, rotation, x and y positions of a sprite, whose dimensions are 1, 3, 6, 40, 32, and 32. The size of the dataset is $737\,280$.

**Data selection** Note that there is no bijection between the factors and the images because of the intrinsic symmetries of the shapes, e.g., $C_4$ of the square and $C_2$ of the ellipse. To this end, we only considered a subset of the original dataset where the orientation only ranges from $0°$ to $90°$, which resulted in a dataset of size $184\,320$. The split of training and test data was similar to the above. Thus, we used only $830/184\,320 \approx 0.5\%$ data for learning augmentations.

**Model and optimization** We used a simple 3-layer MLP ($64 \times 64 \rightarrow 256 \rightarrow 64 \rightarrow$ output) with ReLU activation as the prediction model, cross-entropy (classification) or mean squared error (regression) as the learning objectives, and an Adam optimizer (Kingma & Ba, 2015) with batch size of 32 and learning rate of $1 \times 10^{-4}$.

**Results** Additionally, we show the augmented images in Fig. 9. We can see that these augmentations are not equally easy to learn: the shape and position augmentations perform relatively well, but modifying the scale and orientation may cause shape distortion.

---

[2] 3D Shapes (Burgess & Kim, 2018) `https://github.com/deepmind/3d-shapes` Apache License 2.0

[3] dSprites (Matthey et al., 2017) `https://github.com/deepmind/dsprites-dataset` Apache License 2.0

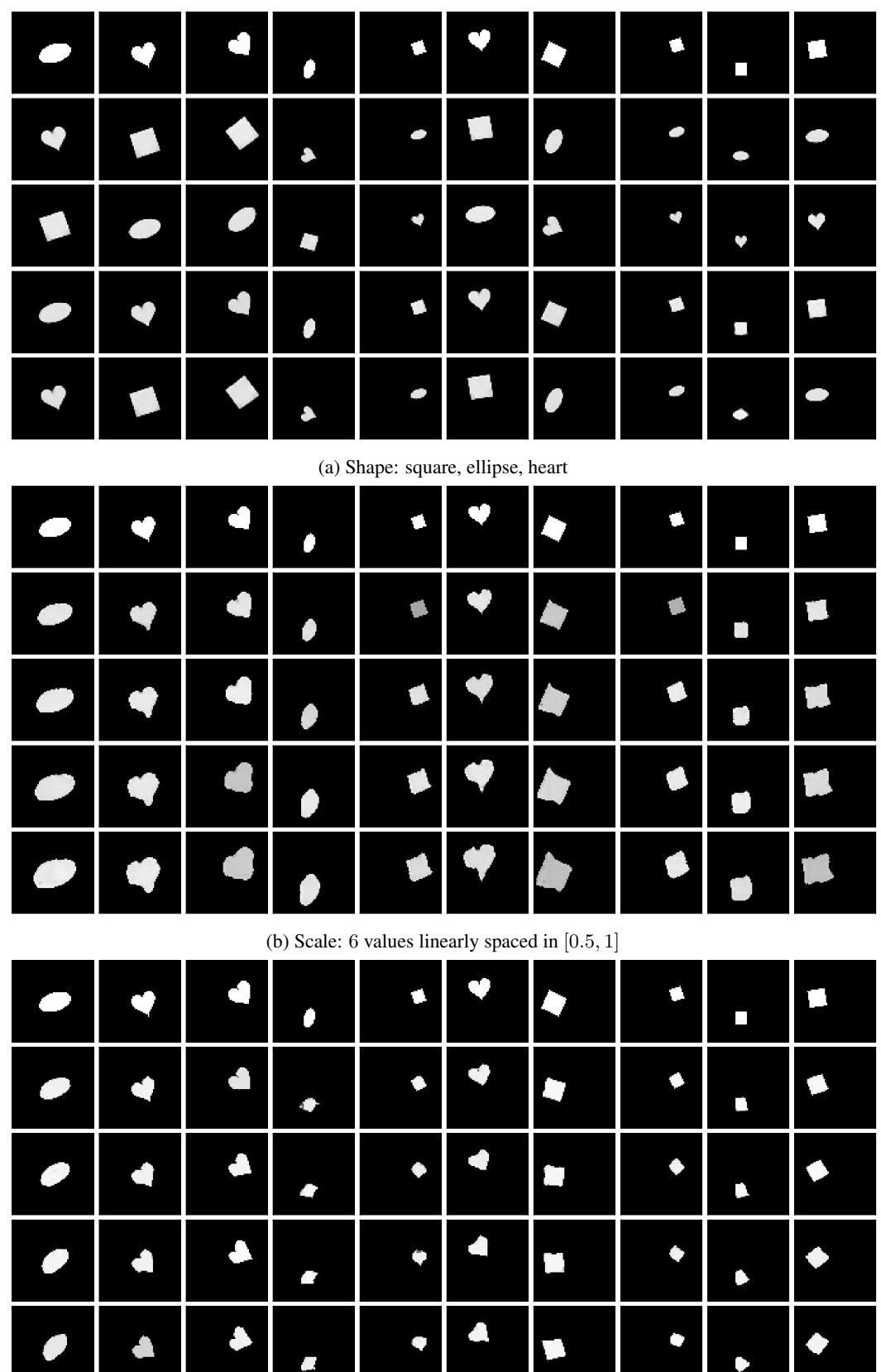

(a) Shape: square, ellipse, heart

(b) Scale: 6 values linearly spaced in $[0.5, 1]$

(c) Orientation: 10 values in $[0°, 90°]$

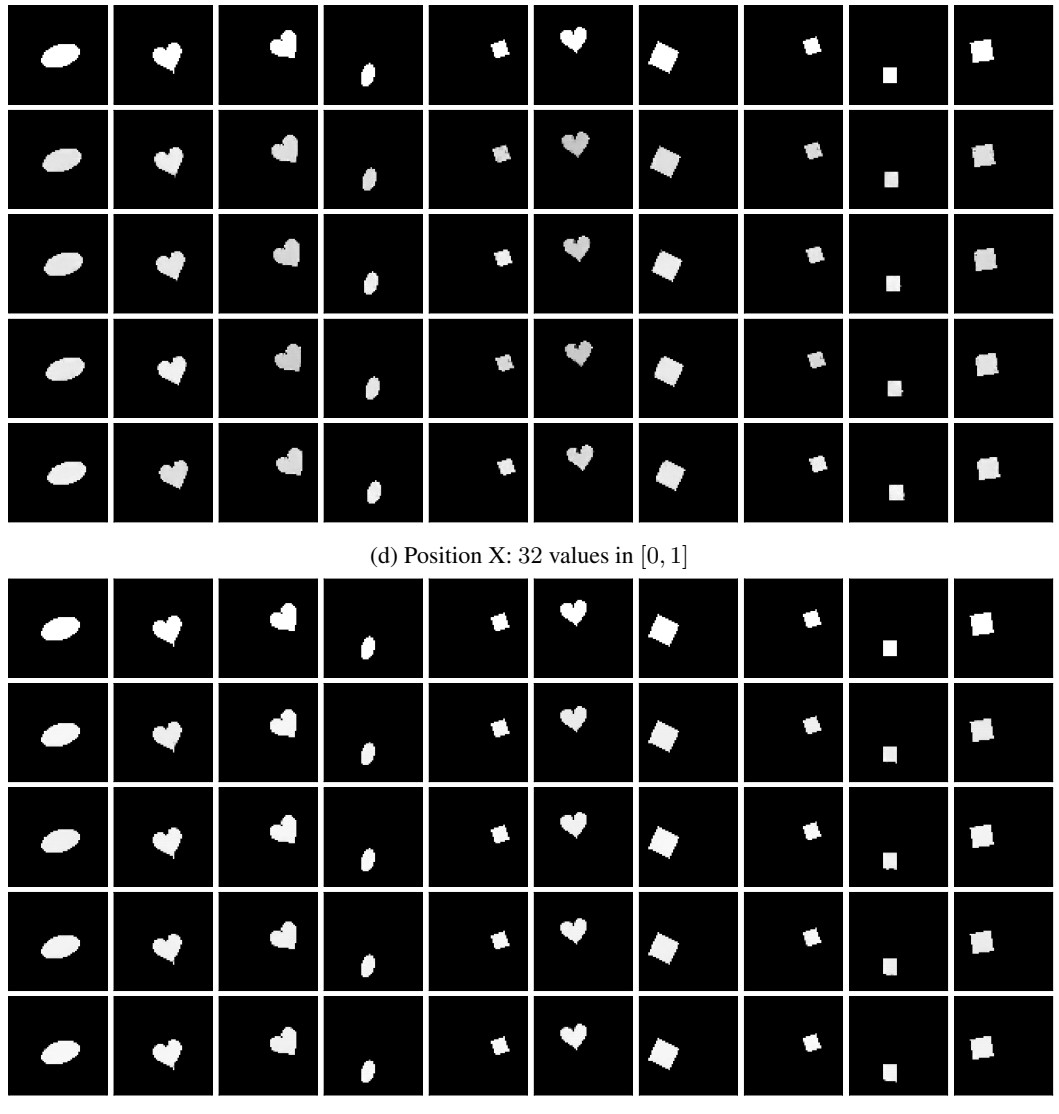

(d) Position X: 32 values in $[0, 1]$

(e) Position Y: 32 values in $[0, 1]$

Figure 9: Augmented training examples of the dSprites dataset

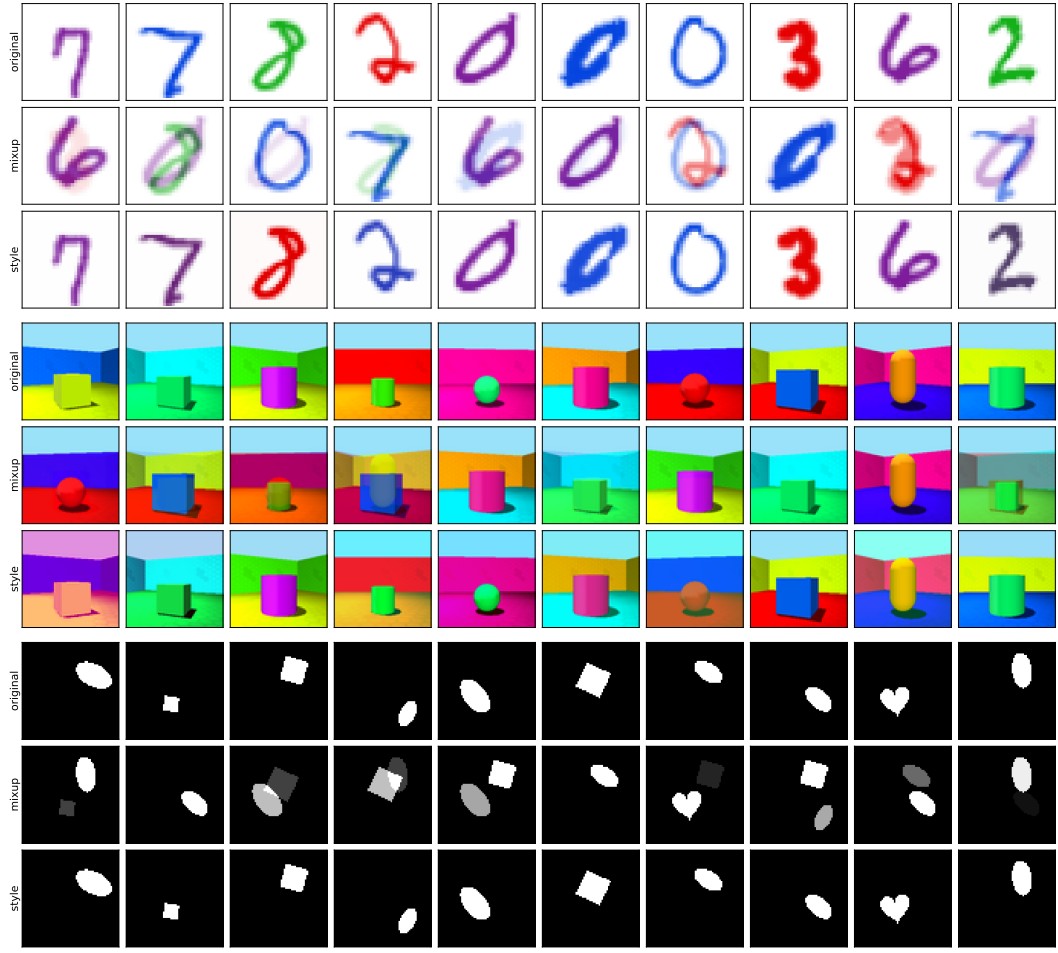

Figure 10: Mixup (Zhang et al., 2018) and MixStyle (Zhou et al., 2021) augmentations on the colored MNIST, 3D Shapes, and dSprites datasets.

## E.4 HEURISTIC AUGMENTATION

Fig. 10 shows the images from the colored MNIST, 3D Shapes, and dSprites datasets augmented by Mixup (Zhang et al., 2018) and MixStyle (Zhou et al., 2021). We can observe that MixStyle actually modifies the colors of the images in the colored MNIST dataset, which may explain why its performance is good in Table 1. Thus, our results also support the claim "heuristic augmentation improves generalization if the augmentation describes an attribute" from the empirical study of Wiles et al. (2022). When it is hard to design augmentations by hand, learning augmentations from data and regularizing these augmentations based on the algebraic constraints is a promising way to improve generalization, which is the main claim of our paper.

