# OpenReview forum: "Equivariant Disentangled Transformation for Domain Generalization under Combination Shift"
_ICLR.cc/2023/Conference — Submitted to ICLR 2023_

### Official Review · Reviewer_fyte · 2022-10-23

**Confidence:** 4
**Correctness:** 3
**Technical Novelty And Significance:** 3
**Empirical Novelty And Significance:** 2
**Recommendation:** 3

**Clarity, Quality, Novelty And Reproducibility:**

- clarity: good
- quality
    - the empirical scope of this paper is probably too limited.
       - Domain generalization has been studied as a popular topic for multiple years recently, and the authors propose a new setting with no particular reason that the existing methods will not work on it, but the empirical scope discards most of those methods.
       - the datasets used in the study seems quite small, thus it's unclear about the true performances of the methods.
   - data selection seems a critical step, it seems unclear how that is done.
- novelty
    - data augmentation operations have Compositionality and Commutativity seems quite intuitive to the community, although this might be an early time of these topics to be formally discussed.
    - another highly relevant work for the prediction phase eq. 10
        - Toward Learning Robust and Invariant Representations with Alignment Regularization and Data Augmentation

**Strength And Weaknesses:**

- strength
    - the setting "combination shift" is reasonably new
    - the algebraic formulation is interesting and might indicate some powerful methods in the future.
- weakness
    - whether the "combination shift" is important needs more discussion. It seems the entire motivation behind this new setting is that this setting is more feasible, which does not seem to be a good reason to study this problem.
    - the empirical scope of this paper is quite limited.

**Summary Of The Paper:**

The paper studies domain generalization, but not in its standard setting, in a new "combination shift" setting instead. The paper also proposes an algebraic formulation for the combination shift problem and proposes some augmentation methods. The paper also has some empirical results.

**Summary Of The Review:**

an interesting discussion, but probably needs further discussions (especially on the empirical end) to be considered for publication.

---

> ### Author Response · Authors · 2022-11-11
> **Responses to Reviewer fyte**
>
>     the authors propose a new setting with no particular reason that the existing methods will not work on it, but the empirical scope discards most of those methods.
>
> Most contents in Section 1 are about the reasons we want and need to solve the combination shift problem and why invariance-based methods may fail. See Table 1 for the performance of existing methods on the combination shift setting.
>
>     data selection seems a critical step, it seems unclear how that is done.
>
> > To train an augmentation $\alpha_i^j$, we need to collect pairs of instances $x$ and $x'$ such that $\mathrm{act}\_X((e_1, \dots, a_i^j, \dots, e_n), x) = x'$, in other words, pairs of the form $x_{y_1, \dots, y_i, \dots, y_n}$ and $x_{y_1, \dots, y_i', \dots, y_n}$, where $\mathrm{act}_{Y_i}(a_i^j, y_i) = y_i'$.

---

### Official Review · Reviewer_CL9J · 2022-10-24

**Confidence:** 3
**Correctness:** 3
**Technical Novelty And Significance:** 2
**Empirical Novelty And Significance:** 2
**Recommendation:** 5

**Clarity, Quality, Novelty And Reproducibility:**

**Clarity and writing suggestions**
- As mentioned before, the current draft of the paper is hard to parse even after the preliminaries discussed by the authors
- In my opinion, it is also unclear if the problem of combination shift and the proposed EDT algorithm can not be explained without algebraic notation.

**Reproducibility**
- Since the paper presents a novel algorithm and experimental results, it would be great if the authors can add a reproducibility statement.

**Strength And Weaknesses:**

**Strength**
- The proposed variant of the domain generalization problem is well-motivated, highly relevant and to the best of my knowledge seems original
- Authors have taken a good effort in setting up preliminaries for algebra used in the paper


**Weakness**
- There are three major weaknesses of the paper:
  -  The writing of the paper is complex and hard to follow for someone who is not well-versed in algebraic formulations. Starting in the abstract and introduction authors use terms like "equivariance", and "disentanglement requirements" without defining what they mean.
  - It is also very unclear if the proposed EDT algorithm follows naturally as claimed in the abstract. While Figure 2 is clear, the cited equations do not naturally seem to follow to me.
  - The experimental validation of EDT is limited to MNIST-like datasets. Authors are encouraged to simulate combination shift settings on Domainbed datasets [1]. Moreover, authors observe modest empirical improvements over MixStyle making it unclear how the proposed augmentation technique will work in other vision datasets.
- Due to the algebraic notation used in the paper, most of the paper is spent on setting up the notation and authors come to the combination shift problem very late in the paper (page 6) which is still mixed with a lot of notations that at times gets hard to keep track of.


[1] Ishaan Gulrajani and David Lopez-Paz. In search of lost domain generalization. In International Conference on Learning Representations, 2021.

**Summary Of The Paper:**

In this paper, authors present a new problem of combination shift (slightly different from the domain generalization problem) where examples from certain domain and label combination pairs are missing in the training data and the hope is to generalize to these examples. The authors discuss this problem with algebraic formulation and present a method that they call equivariant disentangled transformation (EDT). Experimental results highlight issues with some existing techniques and highlight the efficacy of the proposed approach in semi-synthetic settings on several datasets.

**Summary Of The Review:**

Overall, in its current form, the paper is very hard to read even with the preliminaries and setup in Section 3. In particular, it is unclear how EDT is simply followed by their algebraic requirements as claimed in the abstract. Experimental results are limited to small-scale (MNIST-like) datasets with marginal improvements over MixStyle. Authors are encouraged to include results on semi-synthetic in Domainbed datasets.

---

> ### Author Response · Authors · 2022-11-11
> **Responses to Reviewer CL9J**
>
>     Starting in the abstract and introduction authors use terms like "equivariance", and "disentanglement requirements" without defining what they mean.
>
> Thank you for raising your concern. We used terms like [action](https://ncatlab.org/nlab/show/action) and [equivariance](https://en.wikipedia.org/wiki/Equivariant_map) in algebra without adding any new meanings. We discussed the definitions of disentanglement and provided a new definition in this paper. Please see Appendix B for the prerequisites.
>
>     It is also very unclear if the proposed EDT algorithm follows naturally as claimed in the abstract. While Figure 2 is clear, the cited equations do not naturally seem to follow to me.
>
> We claimed in the abstract that `equivariant disentangled transformation (EDT) augments the data based on the algebraic structures of labels and makes the transformation satisfy the equivariance and disentanglement requirements`.  In Fig. 2, Eq. (7) corresponds to component augmentations (single arrows between images on the left). Eq. (8) is actually not shown in this diagram but is illustrated in Fig. 3 (a). Eq. (9) corresponds to the commutative square on the left. Eq. (10) corresponds to commutative squares that contain $f$.
>
>     Moreover, authors observe modest empirical improvements over MixStyle making it unclear how the proposed augmentation technique will work in other vision datasets.
>
> We studied when heuristic augmentations work and do not work. Table 2 shows that MixStyle provides no significant performance gain in some settings because the heuristic augmentation no longer matches the underlying mechanism. See also Appendix E.
>
>     most of the paper is spent on setting up the notation and authors come to the combination shift problem very late in the paper (page 6)
>
> We informally described and illustrated the combination shift problem in Section 1. We defined the problem in Section 2. Later sections are the formulation and our proposed method.

---

> > ### Comment · Reviewer_CL9J · 2022-11-19
> > **Response**
> >
> > I thank the authors for their effort and updates. I read other reviews and I agree with the concerns of Reviewer W6bv. And also due to the lack of empirical validation on the Domainbed benchmark, I stay with my original score.

---

> > > ### Author Response · Authors · 2022-11-21
> > > **Datasets**
> > >
> > > Thanks for your feedback.
> > >
> > > DomainBed is a suite for the domain shift problem. The DomainBed paper questioned the progress of the domain generalization studies and discussed `Are these the right datasets?` (Section 6). Following this direction, we proposed to consider a related but different problem --- combination shift --- which is more realistic and challenging. For this, we provided an algebraic problem formulation and an example of how we can derive useful algorithms from algebraic requirements. To demonstrate the challenges of the new problem setting and show the limitations of existing methods, we need to find appropriate datasets and properly design the experimental settings. At this point, no, datasets in DomainBed do not suit our needs. Please see also [our discussion on the experiments](https://openreview.net/forum?id=bn2J_zqfsEf&noteId=SbMbSzPXHh).

---

### Official Review · Reviewer_W6bv · 2022-11-03

**Confidence:** 4
**Clarity, Quality, Novelty And Reproducibility:** See above.
**Correctness:** 3
**Technical Novelty And Significance:** 3
**Empirical Novelty And Significance:** 3
**Recommendation:** 3

**Strength And Weaknesses:**

Strengths:
1. Domain generalization is a crucial and largely unsolved problem. Any progress on this question could be very impactful.
2. It is sensible to extend Higgins et al's definition of disentanglements from groups to more general algebras.
3. Phrasing invariance algebraically makes sense. The commuting diagrams illustrate the structure very clearly.
4. The proposed regularizers allow for the practical operationalization of the algebraic structures.

Weaknesses:
1. The authors introduce the combination shift problem clearly. But it would be great if they could shed a bit more light on why this is a relevant problem. Are there any real-world problems that fall into this category? Or do they primarily consider this as a stepping stone to solving the full domain generalization problem? In the latter case, could they argue why progress on the combination shift problem should translate into progress on other domain generalization problems?
2. The proposed method requires either knowing the action of domain and label shifts on the data space (act_X), or learning it from data. The authors correctly point out that such data can be learned from paired data of a sample before and after changing the corresponding factors without affecting any other sample details. But the existence of such data pairs is a strong requirement, and this requirement is not part of the problem setup (nor the introduction of the paper). Learning such augmentations from unpaired data is a much harder problem. The authors gloss over this difference around Eq. (7) and in Remark 3, but it is a crucial point. Without strong assumptions, the actions will in general not be identifiable from unpaired data. Therefore I believe that the entire approach hinges on what we could either call inductive bias or wishful thinking: ML algorithms somehow being able to identify act_X even in the absence of theoretical guarantees. Algorithms without theoretical guarantees are of course fine, but the authors could be more explicit about in which sense the algorithm is expected to work.
3. I don't fully understand the experiment in section 5.1. How are the augmentations (act_X) learned here? Could the authors show the effect of the learned act_X somewhere in the appendix?
4. The writing could at times be clearer.

**Summary Of The Paper:**

1. The authors propose to study a problem that they call combination shift: a simplified version of domain generalization in which all environments and labels are available at test time, but not necessarily all combinations.
2. Higgins et al [2018] define disentanglement based on equivariance wrt to the action of a product groups. This paper slightly extends that discussion from groups to more general algebraic structures, in particular semi-groups and monoids.
3. Starting from this definition, the authors argue that one could solve the combination shift problem if the action of domain shifts and label shifts on the input data is known.
4. They point out that this action can be learned from data pairs showing a sample before and after the domain or label has been shifted, but all other aspects are kept constant. They also and discuss learning these actions from unpaired datasets.
5. They demonstrate the approach in experiments.

**Summary Of The Review:**

Formulating disentanglement and domain generalization through algebraic structure is an excellent approach, and there are some good ideas in this paper. However, in its current form I don't think the paper meets the bar. The proposed approach hinges on identifying the (isolated) effect of domain and label shift on data representations from CycleGANs, and there is neither a theoretical guarantee for this to work nor an appropriate discussion of this step. I encourage the authors to discuss this further and to be clearer about their claims.

---

> ### Author Response · Authors · 2022-11-11
> **Responses to Reviewer W6bv**
>
>     combination shift: a simplified version of domain generalization in which all environments and labels are available at test time, but not necessarily all combinations.
>
> Domain shift and combination shift are two different problems of domain generalization; Combination shift is not a simplified version.
>
> All factors are available at training time. We may only need to test a part of domains or classes at test time (Figure 1).
>
>     one could solve the combination shift problem if the action of domain shifts and label shifts on the input data is known.
>
> It is unclear what "the action of domain shifts" refers to here. In our paper, domain shift is a type of domain generalization problem. Action has a specific meaning in algebraic theory (See Appendix B.4).
>
>     In the latter case, could they argue why progress on the combination shift problem should translate into progress on other domain generalization problems?
>
> No, we do not think our proposed method helps solve the domain shift problem. The challenges are different. In our setting, the challenge comes from a limited subset of all combinations and unbalanced distributions of factors. The domain shift problem is hard due to limited knowledge of a completely new domain. Please also see the discussion in [Gulrajani & Lopez-Paz (2021)](https://openreview.net/forum?id=lQdXeXDoWtI) and [Wiles et al. (2022)](https://openreview.net/forum?id=Dl4LetuLdyK).
>
>     But the existence of such data pairs is a strong requirement, and this requirement is not part of the problem setup (nor the introduction of the paper).
>
> It is exactly because paired examples are scarce that we need algebraic regularization (compositionality, commutativity, etc.) to introduce more inductive biases. Section 5.3 and Figure 4 are meant to show this point.
>
>     Learning such augmentations from unpaired data is a much harder problem. The authors gloss over this difference around Eq. (7) and in Remark 3, but it is a crucial point. Without strong assumptions, the actions will in general not be identifiable from unpaired data.
>
> We strongly agree that learning from unpaired data is very challenging. Therefore, we only aimed to solve the challenge of limited combinations instead of unpaired data. We only used labeled data, and we can extract paired training examples for $\ell_0$.
>
>     Could the authors show the effect of the learned act_X somewhere in the appendix?
>
> Augmentation results were shown in Figures 4, 8-10.

---

> > ### Comment · Reviewer_W6bv · 2022-11-11
> > **Response^2**
> >
> > Thanks to the authors for providing a quick response. There were a few instances where I feel like we they did not understand my comments or questions, or I did not understand what they meant. I hope I can clarify in the following.
> >
> > > It is unclear what "the action of domain shifts" refers to here.
> >
> > Does the paper not introduce a monoid that represents label and domain transformations? If I understood correctly, that has a submonoid of label shifts and a submonoid of domain shifts. With "action of domain shift", I mean the action of the domain-shift monoid on the data space, I believe called $\mathrm{act}_X(a, e)$ in the paper.
> >
> > > No, we do not think our proposed method helps solve the domain shift problem
> >
> > I appreciate the authors pointing me to more references. I still wonder if they could mention some real-world problems that have the combination shift structure.
> >
> > > We strongly agree that learning from unpaired data is very challenging. Therefore, we only aimed to solve the challenge of limited combinations instead of unpaired data. We only used labeled data, and we can extract paired training examples for $\ell_0$.
> >
> > I'm afraid I still don't fully understand, and this seems to be a crucial point. What do you mean with "limited combinations"? How do you extract paired training examples – using a CycleGAN-like approach? Could you give me a small concrete example?
> >
> > The optimization problem in Eq. (7) strikes me as underspecified: aren't there many augmentations that could minimize this loss, but are not the "correct" augmentation. Do you claim some sort of identifiability here, or could this approach fail in principle but still works in practice because of inductive biases or other ML magic?
> >
> > As a final, general comment, I appreciate the authors trying to educate their readers and reviewers about algebra. At times this may come across as slightly condescending. I personally like to think I understand the maths well enough, and my issues with the paper are of a different nature.
> >
> > I am looking forward to your responses.

---

> > > ### Author Response · Authors · 2022-11-11
> > > **Clarification**
> > >
> > > > domain shift
> > >
> > > Thank you for the clarification! The term "_shift_" might be ambiguous here because we used _domain/combination shift_ to refer to the problem settings. If you mean $\mathrm{act}_X(a, e)$, it is not known, but it is what we want to obtain -- learned data augmentations.
> > >
> > > > real-world problems
> > >
> > > A classic example is the co-occurrence of objects and backgrounds. For example, the classification results of the CIFAR-10 dataset show that correctly classified bird images usually have a green background (grass/tree), while correctly classified airplane images usually have a blue background (sky). Meanwhile, the classifier makes more mistakes for birds flying in the sky and airplanes landing on the ground. The spurious correlation between object and background exemplifies the combination shift problem. The 3D Shapes dataset we use (wall/floor color vs. object shape) is a proxy for such problems in the real world. Other proxies include the WaterBirds dataset ([Sagawa et al., 2020](https://openreview.net/forum?id=ryxGuJrFvS)), which contains semi-synthetic images of birds with different backgrounds.
> > >
> > > Another example is when the domain contains more than one factor. It is unlikely that we have all combinations of color, texture, position, orientation, light condition, etc., of objects in the training data, while we want the model to generalize to objects with a completely unseen color during the test phase (domain shift). A common case is that we have objects in different conditions, and we want the model to generalize to objects in unseen combinations of conditions (combination shift). The 3D Shapes and dSprites datasets are proxies for such problems.
> > >
> > > > paired training examples
> > >
> > > For example, we want to predict the shape of red/blue triangles/squares. We have red triangles, red squares, and blue triangles (limited combinations of factors, labeled data) during training. A good prediction model should correctly predict the blue squares (unseen combination of factors) during testing.
> > >
> > > We can collect red/blue triangles (paired training examples) and train an augmentation model to change the color from red to blue without changing the shape. We can do so because the colors and shapes are labeled in the training data. _A good color-changing augmentation model trained on triangles should generalize to squares as well_.
> > >
> > > We can also collect red triangles/squares and train a model to change the shape without changing the color. Changing the color and then the shape should get the same result as changing the shape and then the color (commutativity). A good prediction model should reflect such structures in the data space (equivariance).
> > >
> > > From this example, we can see that it is not ML magic, but we do rely on the generalization ability of augmentation models. Currently, there is no guarantee that a color-changing model trained on triangles always works on squares. Therefore, we must carefully choose the labels and design the model space (e.g., NN architectures).
> > >
> > > A bad example is the combination of style/object. A sketch-to-photo augmentation model trained only on animals may fail to generalize to furniture (we may get furry chairs and tables, though!) because some information is missing (colors, textures, etc.). In such a case, our proposed method would fail, and it is better to annotate more factors or obtain more data.
> > >
> > > ---
> > >
> > > Thank you for letting us explain ourselves, and we hope our answers make it clearer.
> > > Please do not hesitate to let us know if you have any remaining questions. Thank you!

---

> > > > ### Comment · Reviewer_W6bv · 2022-11-16
> > > > **Thanks for the clarification**
> > > >
> > > > Thank you for the answer, especially explaining how you learn augmentations.
> > > >
> > > > I can see that this approach works when we have labels that specify a data point fully or to a large extent – like in the case of the colorful shapes.
> > > >
> > > > But why would this work in more complex settings with labels that leave many details open? For instance, consider the case where one environment consists of photographs and one environment of paintings, no further labels. There are many maps on the data space that photographs into paintings on the distribution level, satisfying your Eq. (7). But only one such map (or at least a tiny subset) leaves the content invariant and thus performs a useful augmentation for our purposes.
> > > >
> > > > Of course there are methods for this problem (some of them cited in your paper), but as far as I understand, they don't come with guarantees: their success is purely based on inductive bias and guided by empirical results.
> > > >
> > > > Do you agree with this assessment?
> > > >
> > > > Like I wrote before, I don't think there's anything wrong with guarantee-free algorithms that work in practice. However, I think for such methods a) the empirical demonstration has to be quite convincing and b) this should be very clearly stated.

---

> > > > > ### Author Response · Authors · 2022-11-21
> > > > > **Guarantee**
> > > > >
> > > > > Thanks for your follow-up.
> > > > >
> > > > > ---
> > > > >
> > > > > First, please let us confirm your example. Do you mean:
> > > > >
> > > > > 1. An object classification task;
> > > > > 2. Two environments: photographs and paintings;
> > > > > 3. Labeled environments and unlabeled object classes;
> > > > > 4. Photo-painting augmentations?
> > > > >
> > > > > In such a case, is the task *unsupervised classification*? If so, our proposed method is not applicable because object classes are required to be labeled.
> > > > >
> > > > > ---
> > > > >
> > > > > Second, we understand your concern. Let us consider a slightly different problem where we have three factors (1) object, (2) style, and (3) background. We want to find a map that preserves all three factors. However, only objects and styles may be labeled, but detailed background labels are unavailable. Due to the lack of background labels, we may not obtain background-invariant augmentations.
> > > > >
> > > > > Mathematically speaking, if $\mathbf{A}_i$ are at least monoids (so identity exists), then $\mathbf{A}_1 \times \mathbf{A}_2$ (isomorphic to $\mathbf{A}_1 \times \mathbf{A}_2 \times \mathbf{1}$) can be included into $\mathbf{A}_1 \times \mathbf{A}_2 \times \mathbf{A}_3$ (via $\mathrm{id}: \mathbf{1} \to \mathbf{A}_3$). We can prove that a mapping equivariant to actions of $\mathbf{A}_1 \times \mathbf{A}_2 \times \mathbf{A}_3$ can be equivariant to actions of $\mathbf{A}_1 \times \mathbf{A}_2$ but not necessarily vice versa. Thus, we cannot expect learned augmentations to preserve unannotated factors.
> > > > >
> > > > > Therefore, it is not that algorithms are "*guarantee-free*"; it is *impossible* to guarantee that augmentations satisfy the algebraic requirement of unlabeled factors without further annotations or assumptions (in the same spirit as the impossibility result of [Locatello et al. (2019)](http://proceedings.mlr.press/v97/locatello19a.html)). Our formulation made this clear and precise.
> > > > >
> > > > > Some regularization terms may help in practice. For example, if two trainable functions (NNs) are supposed to be the inverse of each other, it might be easier to obtain the identity functions. However, we found that `when there are more than two factors, cycle consistency alone may not guarantee the identity of non-transformed factors` (Section 5.2), especially when the factor distributions are imbalanced (as in combination shift).
> > > > >
> > > > > ---
> > > > >
> > > > > To contextualize our work:
> > > > >
> > > > > 1. Methods meeting a subset of requirements (only invariance, cycle consistency, etc.) may fail in the new setting;
> > > > > 2. Our method using more algebraic constraints can make use of more supervision signals and works better for labeled factors (but not necessarily for unlabeled factors);
> > > > > 3. Methods guaranteed to satisfy algebraic requirements of unlabeled factors do not exist.
> > > > >
> > > > > Yes, it is important to clearly state what a method can and cannot do. We will further revise the draft and make it clear.

---

### Official Review · Reviewer_ZeBu · 2022-11-04

**Confidence:** 4
**Correctness:** 3
**Technical Novelty And Significance:** 3
**Empirical Novelty And Significance:** 3
**Recommendation:** 6

**Clarity, Quality, Novelty And Reproducibility:**

Clarity - Please see strength 2

Reproducibility - Currently code is not provided

Novelty and Quality - Please see strengths 1,3, 4

**Strength And Weaknesses:**

**Strengths:**
1. The combination shift problem is very well motivated, appropriately formulated and most importantly is an important problem in real world settings
2. Most of the paper (and appendix) is very well written and clear
3. The proposed algorithm (sub section 4.3) is simple, without losing the majorly desired properties
4.  The proof of concept experiments are well designed and show the use case of the proposed approach

**Weaknesses**:
1. While in definition 1, the authors assume that the product algebra in not necessarily finite (via an indexing set), it is unclear  - (the ramifications) of what happens when product decomposition is not countable. The algebraic regularization subsection - maybe is indicative of this - but currently the clarity of that is on the lower side.
2. It is unclear if there is a strategy to learn the product decomposition (without underlying knowledge about the data - and especially in relation to 1 above). Please include synthetic proof of concept experiments to showcase the same.

**Minor:**
1. Is there a citation for the 1st line of the introduction? From cognitive studies/ neuroscience ?
2.  Work [1] which assumes invariance to all groups initially (unless evidence shows otherwise) and then narrows over the subspace lattice  to identify the groups to be not invariant to is relevant work


**References**
1. Mouli, S. Chandra, and Bruno Ribeiro. "Neural Networks for Learning Counterfactual G-Invariances from Single Environments." ICLR 2021

**Summary Of The Paper:**

The authors introduce the combination shift problem towards generalization - especially when the goal is to learn with as few combinations are observed in data. Towards this goal, the author extend the definition of disentaglement beyond of invariances to group actions - highlighting the necessity to exploit the equivariance structure. The authors then provide a strategy to effectively learn data augmentations towards the combination shift problem and improve generalization and subsequently provide proof of concept experiments.

**Summary Of The Review:**

In my opinion, the strengths of the paper out weigh the weaknesses - and therefore recommend border line accept (initial review)

---

> ### Author Response · Authors · 2022-11-11
> **Responses to Reviewer ZeBu**
>
> Thank you for reviewing our work and providing suggestions! We would like to address your questions as follows.
>
>     While in definition 1, the authors assume that the product algebra in not necessarily finite (via an indexing set), it is unclear - (the ramifications) of what happens when product decomposition is not countable.
>
> Thank you for pointing out this issue. Indeed, most algebras we design and use in practice are finite. It is perfectly fine to only use finite components with finite operations:
>
> \begin{equation}
> \\{\mathbf{A}\_i = \\{A\_i, \{f\_i^j: A_i^{n\_j} \to A\_i\\}\_{j=1}^{M\_i}\}\\}\_{i=1}^N
> \end{equation}
>
> where $N$ and $M_i$ are positive integers. However, it excludes some common and useful algebras, such as the additive monoid of natural numbers. Meanwhile, the number of components is usually not large, let alone uncountable, because we want a set of interpretable factors. We use the indexing sets to hint that what is important is the product structure. We will revise this part to reduce confusion (maybe by using a finite set of component algebras whose operations are possibly infinite).
>
>     It is unclear if there is a strategy to learn the product decomposition
>
> No, it is still challenging to learn the decomposition from data now. This task is related to unsupervised/weakly supervised disentangled representation learning.
>
> One focus of disentangled representation learning is extracting underlying factors from unlabeled data. Previous studies mainly use uniformly distributed combinations of factors. In contrast, we focused on the task where the training examples are labeled, but only a small amount of combinations are available.
>
> Ideally, a good model is supposed to extract an appropriate algebra to represent the underlying structure of data from minimal supervision so that it can generalize to unseen combinations of factors well. We are dedicated to achieving this goal.
>
>     Is there a citation for the 1st line of the introduction? From cognitive studies/ neuroscience ?
>
> Thank you for the suggestion. We wrote this sentence mainly based on our experience (as a human). We are aware of one paper, "[_Unsupervised deep learning identifies semantic disentanglement in single inferotemporal face patch neurons_](https://www.nature.com/articles/s41467-021-26751-5)" (Nature Communications 2021), that relates machine learning models with neuroscience in the context of disentanglement. However, it is only limited to the Beta-VAE model. A [NIPS 2017 workshop](https://sites.google.com/view/disentanglenips2017) called for neuroscience or cognitive science insights, but most results seem to be on the machine learning side. We would love to add more support to this sentence if we find appropriate references.

---

> > ### Comment · Reviewer_ZeBu · 2022-12-08
> > **Response to authors**
> >
> > Dear Authors,
> >
> > Thank you very much for the response and the updates. After reading through all other reviews and the raised concerns about evaluation - I will stick to my score(rather than increase).

---

### Author Response · Authors · 2022-11-09
**Conflicting opinions**

We thank all the reviewers for their time and effort in reviewing our work.

We found several conflicting opinions regarding motivation, importance, experiment, and readability. We list inconsistent points of view worth discussing and our explanations below.

---

> ### Author Response · Authors · 2022-11-09
> **Motivation and Importance**
>
>     Reviewer ZeBu: The combination shift problem is very well motivated, appropriately formulated and most importantly is an important problem in real world settings
>     Reviewer CL9J: The proposed variant of the domain generalization problem is well-motivated
>
> vs.
>
>     Reviewer fyte: whether the "combination shift" is important needs more discussion. It seems the entire motivation behind this new setting is that this setting is more feasible, which does not seem to be a good reason to study this problem.
>     Reviewer W6bv: ... it would be great if they could shed a bit more light on why this is a relevant problem. Are there any real-world problems that fall into this category?
>
> We study this problem not just because it is feasible, but because it is more realistic and pervasive. Our claim *"combination shift is a problem worth investigation (at least as much as domain shift)"* is supported by:
>
> - **Previous discussion**: As discussed in [Gulrajani & Lopez-Paz (2021)](https://openreview.net/forum?id=lQdXeXDoWtI) ( See `Are these the right datasets?`  in Section 6) [Wiles et al. (2022)](https://openreview.net/forum?id=Dl4LetuLdyK) (See `We should focus on the cases where we have knowledge about the distribution shift` in Section 5), the domain shift setting may not reflect realistic situations, and other tasks where we have some knowledge of training and test data could be more worth considering. Combination shift is such an example where we have partial information of all domains and classes in the training data.
> - **Other related problems**: We cited and discussed in Section 1 (introduction) and Appendix D (literature review) related settings originated from real-world problems such as *spurious correlation* and *fairness*.
> - **Empirical evidence presented in our work**: Algorithms designed for domain shift based on invariance do not perform well under combination shift, so it is worth considering alternative approaches.
>
> Besides, combination shift could be more challenging than domain shift because we do not assume the training data distributions are balanced, which is reflected in the experiment in Section 5.1.

---

> ### Author Response · Authors · 2022-11-09
> **Experiment**
>
>     Reviewer ZeBu: The proof of concept experiments are well designed and show the use case of the proposed approach
>     Reviewer CL9J: Experimental results highlight issues with some existing techniques and highlight the efficacy of the proposed approach in semi-synthetic settings on several datasets.
>
> vs.
>
>     Reviewer fyte: the empirical scope of this paper is quite limited.
>
> All experimental settings are designed to support our claims:
>
> 1. Section 5.1: We used a simple **MNIST**-based dataset to show that invariance-based methods do not perform well in the combination shift setting, while augmentation-based methods might be more suitable.
> 2. Section 5.2: [Wiles et al. (2022)](https://openreview.net/forum?id=Dl4LetuLdyK) suggested that `Learned data augmentation is effective across different conditions and distribution shifts.` (Takeaway 4 in Section 4.1), but they only used an augmentation method [Goel et al. (2021)](https://openreview.net/forum?id=9YlaeLfuhJF) based on CycleGAN. We pointed out that cycle consistency is only part of the algebraic regularization (Remark 3) and may be insufficient, so we conducted an experiment on the **3D Shapes** dataset (because it has more than two factors) to showcase a failure case (Figure 8). This experiment is also an example of how the algebraic formulation can guide designing data augmentations.
> 3. Section 5.3: We conducted an experiment on the **dSprites** dataset (because of its position factors) to demonstrate the usefulness of the proposed compositionality and commutativity regularizations. This experiment (along with Section 5.2 and Appendix E.4) also shows when and why heuristic augmentation methods such as MixStyle succeed and fail (`In Table 2, we can see that MixStyle provides no significant performance gain in this setting because the heuristic augmentation does not match the underlying mechanism anymore (See also Fig. 10 in Appendix E)`).
>
> If a method does not perform well on a simple dataset, it is highly unlikely that it will achieve better performance on more large-scale datasets. We used simple datasets to show the limitation of existing methods and the potential of the regularized data augmentation approach. Besides, we did not claim in the paper that the proposed method can be applied to real-world problems straightforwardly without any modification. There could be many technical difficulties in practice, whose solutions are, however, out of the scope of our discussion and not our main contribution.
>
> We also did not suggest that neural networks trained via backpropagation are the optimal solution. A manually designed data augmentation may be more data-efficient if we have knowledge about the factors. Program synthesis or diffusion models could be appropriate for more complex datasets, but imposing algebraic regularization on them is a great challenge.
>
> We want to point out that the datasets such as 3D Shapes and dSprites are indeed synthetic, but they are definitely _not small_. For example, dSprites contains **737,280** images, while DomainNet contains 586,575 images. The combination shift is more challenging if the number of combinations is large, so we preferred these datasets.

---

> ### Author Response · Authors · 2022-11-09
> **Readability**
>
>     Reviewer ZeBu: Most of the paper (and appendix) is very well written and clear
>
> vs.
>
>     Reviewer W6bv: The writing could at times be clearer.
>     Reviewer CL9J: The writing of the paper is complex and hard to follow for someone who is not well-versed in algebraic formulations ... the current draft of the paper is hard to parse even after the preliminaries discussed by the authors
>
> We admit that algebraic theory has not been extensively used in the machine learning community (compared with linear algebra, probability, and statistics). It is possible to discuss the regularization in a hand-waving way without the concepts of "equivariance" and "product", but eventually, we will end up *reinventing algebra*. Therefore, as suggested in the abstract and the introduction, understanding some basic algebraic concepts such as monoid/group, action/representation, and equivariance/invariance are **prerequisites** for this work.
>
> We included a brief review in Appendix B, examples in Appendix C, and many diagrams and illustrations to ease readers without such a background. We would appreciate it if the reviewers let us know which part is still confusing. We would like to improve the readability further. Thank you.

---

### Decision · Program_Chairs · 2023-01-20

**Decision:**

Reject

**Justification For Why Not Higher Score:**

The empirical validation is limited to relatively simple datasets and the method is not compared to all relevant previous works, the writing was hard to follow for some of the reviewers, and the method requires knowing the action of domain and label shift, or paired data, which may not be realistic.

**Justification For Why Not Lower Score:**

n/a

**Metareview: Summary, Strengths And Weaknesses:**


The reviewers agree that the problem of combination shift is well motivated and important, but cited a number of limitations of the work in its present form: the empirical validation is limited to relatively simple datasets and the method is not compared to all relevant previous works, the writing was hard to follow for some of the reviewers, and the method requires knowing the action of domain and label shift, or paired data, which may not be realistic. For this reason I recommend to reject the paper in its current form, but encourage the authors to continue working on this important problem and improving the paper.

**Summary Of Ac-Reviewer Meeting:**

n/a